

# El Niño Southern Oscillation (ENSO)-induced hydrological anomalies in central Chile

Renee van Dongen[1,2], Dirk Scherler[1,3], Dadiyorto Wendi[4,5], Eric Deal[6], Luca Mao[7,8], Norbert Marwan[9,10], Claudio I. Meier[11]

[1] GFZ German Research Centre for Geosciences, Earth Surface Geochemistry, Potsdam, Germany.
[2] International Centre for Water Resources and Global Change (UNESCO), German Federal Institute of Hydrology, Koblenz,
Germany.
[3] Freie Universität Berlin, Institute of Geographical Sciences, Berlin, Germany.
[4] GFZ German Research Centre for Geosciences, Hydrology, Potsdam, Germany.
[5] Sunjul GmbH, Berlin, Germany
[6] ETH Zürich, Department of Earth Sciences, Zürich, Switzerland.
[7] University of Lincoln, School of Geography, Lincoln, United Kingdom.
[8] Institute of Geography, Pontificia Universidad Católica de Chile, Chile
[9] Potsdam Institute for Climate Impact Research (PIK), Member of the Leibniz Association, Potsdam, Germany.
[10] University of Potsdam, Institute of Geosciences, Potsdam, Germany
[11] University of Memphis, Department of Civil Engineering, Memphis (TN), United States of America.

*Correspondence to*: Renee van Dongen (vandongen@bafg.de)



**Abstract**

The El Niño Southern Oscillation (ENSO) is a major driver of climatic anomalies around the globe. How these climatic
anomalies translate into hydrological anomalies is important for water resources management, but difficult to predict due to
the non-linear relationship between precipitation and river discharge, and contrasts in hydrological response in regions with
different hydrological regimes. In this study we investigate how ENSO-induced climatic anomalies translate into hydrological
anomalies by focussing on Central Chile (29-42°S), a relatively small area affected by ENSO, that displays steep latitudinal
and elevational climatic gradients. We analyse daily discharge timeseries from 178 discharge stations together with monthly
temperature and precipitation data. Based on the Multivariate ENSO Index (MEI) we classified the discharge data for the time
period 1961-2009 into El Niño (MEI>0.5), La Niña (MEI<-0.5) and non-ENSO periods (-0.5>MEI<0.5), and calculated
relative differences in mean monthly temperature, precipitation, and discharge, as compared to non-ENSO conditions. The
results reveal that precipitation and specific discharge generally increase during El Niño events, while they decrease during La
Niña events. However, there exist large spatial and seasonal variations. The mean monthly precipitation and specific discharge
anomalies during both the El Niño and the La Niña phases are strongest in the semi-arid region (29°-32°S), followed by the
mediterranean (32°-36°S) and humid-temperate (36°-42°S) regions. During El Niño events, the semi-arid and mediterranean
regions experience mean monthly specific discharge increases of up to +396.5% and +104.5%, respectively, and a considerable
increase in the frequency and magnitude of high flows. In contrast, discharge in the humid-temperate region is most sensitive
to rainfall deficits during La Niña events, as revealed by an increased frequency of low flows. We find that the different
hydrological regimes (rainfall- or snow-dominated) show large contrasts in how ENSO-induced climatic anomalies are
translated into hydrological anomalies, in that snowmelt induces a delayed discharge peak during El Niño, provides a minimum
streamflow during dry La Niña conditions, and reduces the discharge variability in rivers. Finally, we discuss the implications
for water resources management, highlighting the need for different ENSO prediction and mitigation strategies in central Chile,
according to catchment hydrological regime.


**Keywords:** El Niño Southern Oscillation (ENSO), central Chile, hydrological anomalies, hydrological response, water
resources management.



## 1. Introduction

Global warming is expected to impact the course of the water cycle, exacerbating hydrological extremes. Global modelling studies predict increasing climate change-related flood hazard risks in large parts of the world (Hirabayashi et al., 2013), whereas other regions are expected to be affected by an intensification of droughts and an increase in the extent of terrestrial drylands (Schlaepfer et al., 2017). Floods and droughts have strong socioeconomic and ecological impacts through their effects on food security (Adams et al., 1999; Iizumi et al., 2014), wild-fires (Beckage et al., 2003; Fasullo et al., 2018; Harrison,

2013), ecosystems (Poveda et al., 2011; Williams and de Vries, 2020), land degradation (Inman and Jenkins, 1999; Morera et al., 2017), and natural hazards (Ward et al., 2014, 2016). Therefore, a better understanding of how climatic anomalies translate into hydrological extremes is important for future water resources and risk management strategies.

In many regions on Earth, one of the main drivers of interannual climatic anomalies, and thus of hydrological extremes, is the El Niño-Southern Oscillation (ENSO) (McPhaden et al., 2006; Salas et al., 2020). ENSO is an oceanic-atmospheric

phenomenon that causes sea surface temperature and wind pattern variations over the tropical Pacific Ocean, resulting in climatic anomalies in South America, Australia, South-East Asia and Africa (Mason and Goddard, 1994; Ropelewski and Halpert, 1987). Central Chile, located omezn the west coast of South America, is one of the regions affected by ENSO. Previous studies of ENSO-induced climatic anomalies in Chile (e.g., Garreaud et al., 2009; Meza, 2013; Montecinos et al., 2000; Montecinos and Aceituno, 2003) identified El Niño as causing a warm and wet phase, while La Niña causes a cold and dry

phase. These climatic anomalies show strong seasonal and spatial variations that are related to the position and intensity of the South Pacific High (Figure 1) (Charrier et al., 2007; Garreaud et al., 2009; Montecinos et al., 2000; Montecinos and Aceituno, 2003).

How ENSO-induced climatic anomalies translate into hydrological anomalies and in particular hydrological extremes is not easily assessed, due to the non-linear relationship between precipitation and river discharge (Stephens et al., 2015), and the

multiple effects of air temperature on the water holding capacity of the atmosphere, evapotranspiration, and snowmelt (Emerton et al., 2017, 2019; De Perez et al., 2017). Furthermore, a basin's sensitivity to climatic anomalies is also controlled by catchment characteristics such as catchment area, elevation, bedrock lithology, regolith thickness, and vegetation cover (e.g., Post and Jakeman, 1996; Rust et al., 2020). Asymmetric streamflow responses to ENSO-induced precipitation anomalies have been observed in some river basins around the world, induced by snow cover, soil moisture, or the integration of

cumulative hydrological processes in large basins, whereas other catchments display a symmetric response to climate anomalies (Lee et al., 2018; Mosley, 2000). A recent study of climate change effects on river floods in Europe revealed substantial regional variations in such extreme events (Blöschl et al., 2019; Kemter et al., 2020). Although not focussing on ENSO-induced climatic anomalies specifically, this study demonstrates the need for analyses at high spatial resolution to capture the spatial complexity of hydrological response to climate.

Central Chile (29°- 42°S) features a strong latitudinal climatic gradient, from semi-arid conditions in the north to a humid-temperate climate in the south (Pizarro et al., 2012; Valdés-Pineda et al., 2016), as well as large elevation differences between the high-elevation Andes mountain range in the east and the low-elevation coastal region in the west. Due to these altitudinal contrasts, the Andes and the coastal region are characterized by different hydrological regimes. Andean catchments in the northern part of central Chile experience their main discharge peak during the glacier- and snowmelt season in summer (nival-

type), whereas river discharge peaks during the winter rainy season for basins in the coastal region (pluvial-type) (Alvarez-Garreton et al., 2021; Oertel et al., 2020). Given the strong latitudinal and altitudinal gradients that occur over a relatively small area, central Chile, with its relatively dense hydrometeorological network, is well-suited for a detailed investigation of how ENSO-induced climatic anomalies are converted into hydrological anomalies.

ENSO-induced hydrological anomalies likely cause relevant socioeconomic impacts in Central Chile, as the strongest climatic

anomalies occur in the semi-arid and mediterranean regions (~29°-36°S), where the majority (~60%) of the population resides (Valdés-Pineda et al., 2014). These regions are strongly dependent on fresh water for crop irrigation, domestic water use, and





hydropower generation (Alvarez-Garreton et al., 2018; Cordero et al., 2019; Masiokas et al., 2006; Meza, 2005). A detailed understanding of how ENSO induces hydrological anomalies is therefore crucial for water resources management in central Chile.

Previous studies have investigated the effect of ENSO in central Chile on precipitation and temperature anomalies (e.g., Garreaud et al., 2009; Hernandez et al., 2022; Montecinos et al., 2000; Montecinos and Aceituno, 2003; Ropelewski and Halpert, 1987), snow cover (e.g., Cordero et al., 2019; Cortés and Margulis, 2017; Masiokas et al., 2006), and river discharge (Hernandez et al., 2022; Piechota et al., 1995; Rubio-Álvarez and McPhee, 2010; Waylen et al., 1993; Waylen and Caviedes, 1990). Generally, these studies reported enhanced precipitation and river discharge during the warm and wet El Niño phase,

and reduced precipitation and river discharge during the cold and dry La Niña phase, but the reported climatic and hydrological anomalies suggest strong seasonal and spatial deviations. The studies of Waylen et al. (1993) and Oertel et al. (2020) compared streamflow and precipitation data to investigate the differences in response to ENSO. Both studies reported a time lag between the precipitation and streamflow responses and attributed this to snow cover. However, these studies focussed predominantly on Andean basins and included relatively sparse coverage of river basins (15 and 20 river basins, respectively, over a latitudinal

distance of ~ 1200 km). In a recent study Hernandez et al. (2022) investigated the controls of both ENSO-induced temperature and precipitation anomalies on streamflow in 59 near-natural catchments between 30°S and 42°S. This study concluded that ENSO exerts a significant control on the climatic and hydrological variability in the region. Streamflow variability is primarily driven by precipitation variability and secondarily by late spring and winter temperatures. Even though this study is based on a larger dataset of 59 stations, the catchments are predominantly located in the Andes. Furthermore, this study does not focus

on hydrological extremes and evaluates streamflow responses on a yearly basis, whereas a seasonal focus would be advantageous to discover temporal shifts in ENSO responses.

To date, a high-resolution study on the spatial and temporal differences in ENSO-induced climatic and hydrological anomalies, which differentiates between the Andes and the coastal region is still lacking. Accordingly, in this study we investigate how ENSO-induced climatic anomalies translate into hydrological anomalies across central Chile (29°-42°S), by analysing quality-

controlled daily discharge data from 178 stations (Alvarez-Garreton et al., 2018). We focus on ENSO-induced mean monthly temperatures and precipitation anomalies, and how these are converted into mean specific discharge anomalies as well as in differences in the frequency of low- and high-flow regimes. Finally, we discuss the implications of our results for water resources management.

## 2.   Climate in central Chile

Central Chile (29°-42°S) covers a north-to-south climatic gradient ranging from cold semi-arid and cold desert climates (BSk) in the northern-central region (29°-32°S), across a subhumid mediterranean climate (Csb) in the central region (32°-36°S), to humid mediterranean and temperate rain-oceanic climates (Cfb) in the southern-central region (36°- 42°S) (Köppen, 1900; Kottek et al., 2006). For brevity, we use from now on the simpler terms semi-arid (29°- 32°S), mediterranean (32°- 36°S), and humid-temperate (36°- 42°S) (Figure 2). Besides the north-south contrast in climate, the elevation differences between the low

elevation coastal region and the high elevation Andes Mountain range, located approximately 100 km farther to the east, promote an additional east-west climatic gradient in temperature and precipitation. On average, the orographic forcing of precipitation by the Andes accounts for a two to three times higher precipitation rate in the mountains as compared to the coastal region, at any given latitude (Barrett and Hameed, 2017; Garreaud et al., 2009; Viale and Garreaud, 2014).

Most of the seasonal and interannual climatic variability in Chile is controlled by the South Pacific High (SPH), a semi-

permanent anticyclone centred over the south-eastern Pacific Ocean (e.g., Barrett and Hameed, 2017; Montecinos and Aceituno, 2003; Schneider et al., 2017). The position of the SPH determines the boundary between the arid region in the north, which is affected by the subsidence of dry air in the Hadley cell (quasi-barotropic conditions), and the humid region in the





south, under the influence of the westerlies (baroclinic conditions). Therefore, the seasonal migration of the SPH and thus, the boundary between quasi-barotropic and baroclinic conditions, plays a crucial role in the seasonal variations of temperature and

precipitation. During winter, when the SPH is located at its most north-eastern position (26-30°S, 85-95°W) (Figure 1a), extratropical cyclones associated with the westerlies have their largest meridional extent (Fuenzalida, 1982; Montecinos and Aceituno, 2003), resulting in wet conditions in the humid and mediterranean regions (Figure 1b). During summer, most of central Chile experiences dry and warm conditions, when the SPH is located at its most south-western position (33-36°S, 100-108°W) (Ancapichún and Garcés-Vargas, 2015; Montecinos and Aceituno, 2003). Under such conditions, dry air from the

Hadley cell subsides over a large meridional extent (Dima and Wallace, 2003), causing low precipitation rates in the semi-arid and mediterranean regions of Chile (Figure 1b).

The position and intensity of the SPH plays an equally important role for ENSO-induced precipitation anomalies (Barrett and Hameed, 2017; Kiladis and Diaz, 1989; Montecinos et al., 2000; Montecinos and Aceituno, 2003). In South America, El Niño conditions develop from pressure differences above the Pacific Ocean that weaken or reverse the equatorial trade winds,

pushing warm sea surface waters from the western Pacific Ocean towards the west coast of South America (Jaksic, 1998; Ropelewski and Halpert, 1987). Due to the anomalously warm sea surface temperatures near the coast of South America, this state is also termed the warm phase of an ENSO event. La Niña, or the cold phase, occurs when the system changes to a reinforced condition of the normal state, in which the Humboldt Current brings cold water along the west coast of South America, which then flows towards the eastern Pacific Ocean. Typically, this cold phase happens after an El Niño phase (Diaz

and Kiladis, 1995).

During El Niño phases the SPH intensity weakens, which results in the blocking of storm tracks across the Amundsen-Bellinghausen Sea and the intensification of the westerlies at mid-latitudes, leading to wetter conditions over central Chile (Garreaud and Battisti, 1999; Montecinos and Aceituno, 2003; Rutllant and Fuenzalida, 1991). During La Niña phases, strengthening of the SPH drives a southward displacement of midlatitude storm tracks, resulting in a larger region to experience

dry conditions under the influence of the Hadley Cell (Montecinos and Aceituno, 2003). As the boundary between the areas under quasi-barotropic and baroclinic conditions follows a seasonal north-south movement, controlled by the position of the SPH, climatic anomalies are found to predominantly concentrate on the semi-arid region during winter, when this boundary is at its northern-most position, whereas they centre on the humid-temperate region during spring, when this boundary is located further south (Montecinos and Aceituno, 2003). When ENSO conditions prevail, either El Niño or La Niña, the SPH changes

its intensity in the summer season. This induces different climatic anomalies during El Niño and La Niña summers, as compared to the other seasons under ENSO phases (Garreaud et al., 2009; Montecinos et al., 2000; Montecinos and Aceituno, 2003).

Several studies have noted reduced correlations between ENSO indices and streamflow from the 1920s to about 1950, for South America (e.g., Dettinger et al., 2000; Elliott and Angell, 1988). This has been attributed to a reduced ENSO intensity

and presumably weaker oceanic-atmospheric teleconnections (Dettinger et al., 2000). Furthermore, since 2010, Chile has been affected by a long-lasting megadrought, with 25-45% reductions in precipitation (Alvarez-Garreton et al., 2021; Boisier et al., 2016; Garreaud et al., 2017, 2020). Recent studies suggest that this megadrought is not related to ENSO (Garreaud et al., 2020), but is instead partly induced by the Pacific Decadal Oscillation and, to a smaller extent, by anthropogenic climate change that affected the ozone layer (Boisier et al., 2016, 2018a). To exclude any influence of this megadrought on the results

of this study, we did not consider any data after 2009. To ensure for an adequate, coherent comparison between river discharge, precipitation and temperature data, we decided to focus on the time period 1961-2009 in this study, for which high-quality precipitation and temperature data is available (see section 3.2).





### 3. Methodology

#### 3.1. Daily river discharge data

The study is based on the Catchment Attributes and MEtereology for Large sample Studies for Chile (CAMELS-CL) dataset, which includes 516 river discharge stations across Chile (17.8°-55.0°S). The data is obtained by the Dirección General de Aguas (DGA; part of the Public Works Ministry) and provided by the Center for Climate and Resilience Research (CR²). CAMELS-CL is a quality-checked meteo-hydrological dataset that includes daily river discharge time series from stream gauging stations and daily meteorological timeseries (temperature, precipitation, potential evapotranspiration, and snow water

equivalent) derived from gridded data products (Alvarez-Garreton et al., 2018). Additionally, the dataset provides information on catchment attributes such as human intervention levels, land cover, and the presence of large dams.

We focus on the time period from 1961 to 2009, for which both discharge data in the CAMELS-CL dataset and the Multivariate ENSO Index data (MEI; Wolter and Timlin, 1993, 1998, 2011) – on which we based the ENSO-phase classification – exist. From the 516 stations in the CAMELS-CL dataset, we selected 178 catchments for further analysis (Table S 1), based on the

following requirements. First, we selected stations based on data availability during the chosen time period (1961-2009), requiring a record length of at least 10 years (Figure S 1a, b). We furthermore excluded catchments with large hydropower or irrigation dams (Figure S 1c), as well as basins with >10% of anthropogenic water extraction as compared to the annual discharge (Figure S 1d). The latter quantity is called the human intervention degree and is computed from the annual volume of granted water extraction rights (provided by the National Water Atlas; DGA, 2016), expressed as a flow rate, normalized

by mean annual river discharge (Alvarez-Garreton et al., 2018). Unfortunately, information on the actual extraction volumes is currently lacking (Alvarez-Garreton et al., 2018). Based on a 30-m resolution land cover dataset provided for Chile (Zhao et al., 2016), we furthermore excluded catchments for which >5% of the surface area is covered with impermeable surfaces (e.g., cities) (Figure S 1Figure S 1e), as well as catchments where >50% of the area is used for agriculture (Figure S 1f). Finally, we excluded catchments which are located directly downstream of large (>10% of catchment area) lakes (Figure S

1Figure S 1g), because these regulate discharge due to their large storage capacity.

For the semi-arid and mediterranean climate zones (29°-36°S), many of the coastal region catchments had to be removed due to the presence of dams or considerable human water extraction. Increasing the threshold for the human intervention degree did not increase the number of such catchments, likely because the occurrence of water extraction correlates with the presence of dams, and extensive water extraction volumes for irrigation and the mining industry are common in these regions (Aitken

et al., 2016). Hence, the majority of the 178 catchments used in this study are located in the humid-temperate region (36°-44°S) (Figure S 1h). After all filtering, we classified the river catchments into Andean (high elevation) or coastal region (low elevation) river catchments (Figure S 1h). We could not classify the catchments based on an elevation-threshold, because the Andes decrease in elevation towards the south. Instead, we derived a hillslope map from a 90-m digital elevation model (DEM) provided by the Shuttle Radar Topography Mission (SRTM; Jarvis et al., 2008), and created a boundary based on increasing

hillslope angles at the Andean foot slopes (red line in Figure 2a). Some large river basins drain both the Andes and coastal region; these were classified according to the region where the majority (>50% of the catchment area) of the basin is located. After this classification, the final data set consists of 99 Andean and 79 coastal region catchments (Figure 2a and Table S 1).

At most stations, the daily discharge records contain missing values, with gap lengths that vary from a single day up to several months (Figure 2c). To reduce adverse effects of data gaps on our analysis, we further processed the discharge records in the

following order (Figure S 2). First, we removed all months containing data gaps of >10 days (Figure S 2a). Second, we determined specific, tolerable gap lengths for each station, based on a maximum-tolerated lag length for which the Pearson autocorrelation function (ACF) coefficient r of the mean daily flows is above 0.7 (and p-value < 0.001) (Figure S 2b and c). Discharge data gaps with lengths below the specific tolerable gap length threshold were linearly interpolated. With this approach we assume that the minimum ACF value of 0.7 is a safe indicator to perform the linear interpolation. In the following



analysis, we only used months that did not have any remaining daily data gaps, after applying the above steps (Figure S 2d). Finally, we computed mean monthly specific discharge (MMQsp) by normalizing the discharge by upstream catchment area, as reported in the metadata. This facilitates direct comparison between runoff and precipitation data, as they have the same units (mm day$^{-1}$).

### 3.2. Precipitation and temperature data

The CAMELS-CL dataset also includes mean daily precipitation and temperature data from various data providers (Alvarez-Garreton et al., 2018). Amongst those is the CR2MET dataset, which is a 0.05° gridded historical climate data product, specifically developed for Chile, and calibrated and validated with a large network of climate stations (Alvarez-Garreton et al., 2018; Boisier et al., 2018b). However, we did not use this dataset for our analysis of ENSO effects, as it only covers the period starting 1979. For more comprehensive comparison between ENSO-induced temperature, precipitation, and specific discharge anomalies, we sought climatic data with a longer temporal overlap with the river discharge data. Because of the monthly resolution of the Multivariate ENSO index, a monthly climatic product was deemed sufficient for such purposes.

We tested the quality of the 0.25° resolution mean monthly precipitation (MMP) dataset from the Global Precipitation Prediction Centre (GPCC; Meyer-Christoffer et al., 2015) which provides data from 1950 onwards, and a 2.5-minutes resolution dataset from CRU-TS-4.03, downscaled with WorldClim V2.1 (Fick and Hijmans, 2017; Harris et al., 2014) which provides MMP starting in 1961. We compared MMP from both datasets to the MMP data from the CR2MET dataset, for the time period 1979-2009. Based on monthly correlation coefficients of MMP in the upstream area of each station, and assuming the CR2MET dataset to be closest to reality, we concluded that the WorldClim V2.1 was the best performing MMP dataset for the majority of the catchments (Figure S 3) and selected this dataset for our analysis. However, compared to the CR2MET dataset, the WorldClim V2.1 dataset was found to overestimate precipitation (by up to ~100 mm) in steep Andean catchments in the mediterranean and humid-temperate regions (>32°S) during the snow season from April to September (Figure S 4). Furthermore, we selected the 0.50° resolution mean monthly surface air temperature (MMT) dataset from the Climate Prediction Centre (CPC; Fan and van den Dool, 2008), which provides MMT time series starting in 1948. Comparing the CPC and CR2MET datasets revealed that the CPC dataset underestimates land-measured mean monthly temperatures during the summer (average -2.34°C) and autumn (average -2.45°C) seasons in the mediterranean and humid-temperate regions and year-round in the semi-arid region (average -5.95 °C) (Figure S 5).

Despite the underestimation of precipitation and temperature in the WorldClim V2.1 and CPC datasets, respectively, as compared to the CR2MET, we decided to use these datasets in our study. Due to their data coverage for the 1961-2009 period, they allow for direct comparison between ENSO-induced differences in MMT, MMP and MMQsp. We did not conduct a bias correction, because the bias was different for the Andes vs. the coastal region and the different climate zones. Therefore, it was impossible to find one common formula to correct all the data with. It is further worth noting that in the following analyses, we only used MMP and MMT values for those months with discharge data, at each station. To obtain for each of the 178 discharge stations basin-averaged MMP and MMT timeseries for the study period (1961-2009), we first derived upstream drainage areas using the SRTM DEM and the TopoToolbox v2 (Schwanghart and Scherler, 2014). We then derived MMP and MMT values using nearest-neighbour interpolation for each DEM grid cell in the drainage areas, and calculated arithmetic means.

### 3.3. ENSO-classification

The ENSO-phase classification in this study is based on the value of the original Multivariate ENSO Index (MEI; Wolter and Timlin, 1993, 1998). We used the original MEI data, as the recently released MEI.v2 data only provides values of the index starting in 1979. Following the method suggested by the MEI developers, we compared the two-month-averaged MEI index




(month(i) and month(i-1)) to the climatic and hydrological data of month(i). This is advised because the atmospheric response to tropical sea surface temperature anomalies shows a time lag (Wolter, 2018).

To preclude any potential effects of the delayed atmospheric response to ENSO, we conservatively classified climatic and hydrological data in different ENSO classes by only including data from longer-lasting ENSO-phases. We defined the thresholds for El Niño events (MEI > 0.5), La Niña events (MEI < –0.5), and non-ENSO periods (–0.5 < MEI < 0.5), and only

classified data in the different ENSO classes when the threshold criterion was met for three or more consecutive months in a row. Months that did not meet these criteria were discarded from the analysis. Out of 588 months during the time period 1961-2009, this approach resulted in the classification of 149 months as El Niño, 113 months as La Niña, and 136 months as non-ENSO (Figure 2b). A total of 190 months were not classified in any of the ENSO-classes, because MEI values were in a transition period between different ENSO phases and were not sufficiently stable for at least three consecutive months. During

the studied time period, the distribution of El Niño events, La Niña events, and non-ENSO periods is not evenly distributed over the seasons of the year. El Niño phases occur less often in the autumn and winter months, whereas La Niña phases are slightly less frequent during the winter season (Figure S 5). Non-ENSO phases are relatively equally distributed over the four seasons, with a slightly higher frequency during spring. Over the 1961-2009 time period, non-ENSO periods are relatively evenly distributed (Figure S 6), but El Niño events occurred more often after 1975 while La Niña events occurred more often

before 1975.

### 3.4. Data analysis

We investigated ENSO-related differences in climate and river discharge for the entire year, as well as the four seasons. We classified the months into Austral seasons: summer – December, January, February; autumn – March, April, May; winter – June, July, August, and spring – September, October, November. Based on the seasonality of precipitation, snowmelt, and

runoff, the hydrological year is defined from autumn to summer (Alvarez-Garreton et al., 2018), which is also the order of seasons shown in the figures of this paper.

For stations and months with available discharge data, we classified MMQsp, the CPC-derived MMT, and WorldClim V2.1-derived MMP data in different ENSO-phases classes based on MEI values, as explained in Section 3.3. We further classified the data within each ENSO-class according to the above-defined seasons. If a seasonal ENSO-class contained sufficient data

(>10 months), we calculated long-term average MMT, MMP, and MMQsp values for each season. To additionally study annual differences between El Niño and La Niña events, compared to non-ENSO phases, we also calculated the mean annual temperature (MAT), mean annual precipitation (MAP), and mean annual specific discharge (MAQsp) for each station. To avoid biases towards a certain season that occurs more frequently in a particular ENSO-phase, we calculated MAT, MAP, and MAQsp based on the arithmetic mean of the long-term monthly values for each season. Next, we computed the relative

differences in MMP, MMT, and MMQsp of El Niño and La Niña phases versus non-ENSO phases, which we present as percent deviations from non-ENSO conditions. We summarized the percent differences in MMT, MMP, and MMQsp for each climatic and elevation (coastal region or Andes) region in boxplots, and reported the differences based on the median values of all stations in each region and the 5[th]- and 95[th]-percentile values.

Furthermore, we investigated whether the frequency of high flows and low flows change during El Niño and La Niña phases,

as compared to non-ENSO conditions, which provides important information for flood and drought risk management (Figure 3a). For each station, we first defined low- and high-flow thresholds based on the 5[th]-percentile ($Q_5$) and 95[th]-percentile ($Q_{95}$) values of the empirical distribution of the non-ENSO daily specific discharges. For each of the three cases (El Niño, La Niña, and non-ENSO conditions) we next calculated two areas under the corresponding empirical probability density function (pdf) of daily specific discharges: the area below the low-flow ($Q_5$) threshold, as well as that above the high-flow ($Q_{95}$) threshold.

For simplicity, we termed these areas the quantile areas (QA) (Figure 3a). The quantile area above the high-flow threshold ($QA_{95}$) reflects the frequency and magnitude of the high-flow regime, whereas the quantile area below the low-flow threshold




($QA_5$) reflects the frequency and magnitude of the low-flow regime. Finally, we calculated the relative differences in each quantile area during El Niño and La Niña events, as the percent difference compared to non-ENSO periods ($\Delta QA$). Changes in QA can occur due to changes in both the frequency and magnitude of low-flow and high-flow events. It is worth noticing
here that although we use the terminology "high flow" and "low flow" for this analysis, their association to floods and droughts is not necessarily straightforward nor robust. A high-flow regime defined by $Q_{95}$ cannot directly be considered a flood, although the $Q_{95}$-threshold has often been used in other studies to define floods. The same applies to droughts that are not necessarily directly linked to our low-flow regime based on $Q_5$. Droughts could instead be defined using more complex notions, such as the effective drought, which could be defined as water scarcity for a certain, defined period of days to weeks, depending on
the region and basin size. However, as such scores and definitions would require additional data, we decided to adopt the terminology high-flow and low-flow regime instead.

Differences in the quantile areas ($\Delta QA$) during El Niño and La Niña events can be the result of either 1) changes in the mean specific discharge (i.e., a shift of the entire empirical distribution of daily specific discharge towards higher or lower discharge magnitudes) or 2) changes in the shape of the distribution, i.e., when the left or right tails of the distribution become lighter or
heavier (Figure S 7). A change in the shape of the specific discharge distribution would occur if ENSO differentially affects certain ranges of specific discharge more strongly than others. For both the process understanding of the hydrological response to ENSO and the context of water resources management, it is important to know the range of specific discharge magnitudes that are affected. Hence, we additionally investigated the variability of daily specific discharge (i.e., how often and by how much discharge deviates from the mean discharge), and how that differs between El Niño, La Niña, and non-ENSO phases.
To study discharge variability, previous studies have parameterized the shape of the magnitude-frequency distribution of daily discharge with various distributions (e.g., Pareto (Molnar et al., 2006), inverse gamma (Lague et al., 2005), and stretched exponential (Rossi et al., 2016)). A study in a region with distinct low- and high-flow seasons fitted the high-flow and low-flow regimes with the weighted sum of two inverse gamma distributions (Scherler et al., 2017). The inverse gamma distribution combines an exponential tail for the low-flow regime with a power law distribution for the high-flow regime, and better fits
the roll-over from lower towards higher discharge magnitudes compared to, e.g., a Pareto distribution (Lague et al., 2005). Therefore, we decided to use the inverse gamma distribution to investigate differences in the shape of daily discharge distributions when El Niño or La Niña events occur, as compared to non-ENSO periods. We tested the use of both a single and the weighted sum of two inverse gamma fits but decided for a single inverse gamma fit, due to the limited amount of daily discharge data when the data are divided across the different ENSO phases and seasons. Fitting an inverse gamma distribution
yields the shape parameter 'k', which characterises the relationship between mean discharge and extreme events and can be interpreted as a measure of the discharge variability. Low k values are associated with high variability, while high k values are associated with low variability. In our analysis of daily discharge variability, we only included stations with reliable fits ($R^2>0.97$) (Figure 3b). Again, we calculated the relative differences in the k-parameter ($\Delta k$) for the two ENSO stages (El Niño, La Niña) as their percent differences compared to non-ENSO conditions (Figure 3c).
We calculated both $\Delta QA$ and $\Delta k$ for each individual season, as well as for the annual data. Results from the two different methods, for both El Niño and La Niña phases, are presented as individual figures showing all stations in the data supplement, with the mean $\Delta QA$ and $\Delta k$ for all stations located within 1° latitudinal window (solid line), as well as the 1 σ-standard deviation (shaded background) (Figure 3c). In the following, we combined the $\Delta QA$ and $\Delta k$ results for each 1°-latitudinal window into a single plot (Figure 3d). In this figure, the marker values and error bars on the x-axis represent the mean percent
difference and 1σ-standard deviation in $\Delta QA$, respectively, for all stations within a 1°-latitudinal window, while the marker colour depicts the mean $\Delta k$ value over that latitudinal window. Figure 3d directly shows whether any differences in either the high-flow or the low-flow regimes occur during El Niño or La Niña events, as compared to non-ENSO periods ($\Delta QA$, x-axis), and whether these are related to a change in shape of the daily discharge distribution ($\Delta k$, colour-coding).



## 4. Results

In this chapter we report the differences in mean monthly temperature (MMT), mean monthly precipitation (MMP), mean monthly specific discharge (MMQsp), and the frequency of low flows and high flows during El Niño and La Niña events, as compared to non-ENSO periods. We analyse all data on both yearly and seasonal bases. Furthermore, we present the differences across the three climatic regions: semi-arid (29-32°S), mediterranean (32-36°S) and humid-temperate (36-42°S), as well as the contrasts between the Andes and the coastal regions. For all combinations of regions and seasons, including the

annual case, we report median values of these differences, corresponding to the median of all station values in that region, over that season. Furthermore, we report the 5th and 95th percentile values for each region and season ($T_5$ and $T_{95}$ for temperature, $P_5$ and $P_{95}$ for precipitation, and $Q_5$ and $Q_{95}$ for discharge).

### 4.1. El Niño

#### 4.1.1. Differences in mean monthly temperature (MMT) and mean monthly precipitation (MMP)

Compared to non-ENSO phases, the MMT is higher in about 88% of the studied river catchments during El Niño events, but the magnitude of the MMT increase is rather small (Figure 4 and Table S 2). Most of the catchments that show lower temperatures during El Niño are located in the semi-arid region, where we also observe large variations in temperature anomalies, with temperature decreases of up to -104.0% ($T_5$) and increases of up to +253.2% ($T_{95}$) (Table S 3). The largest temperature increases when averaging across all regions are found in the autumn and winter seasons, with median temperature

($T_{50}$) increases of +8.4% and +8.5%, respectively (Table S 3). During these seasons the temperatures accrue by up to +122.8% and +74.3% ($T_{95}$), respectively. During the spring and summer seasons, many catchments do not experience large temperature differences with $T_{50}$ increases of only +0.9% and +0.5%, respectively (Table S 3). Finally, the annual temperature increase is higher in the Andes with a median temperature ($T_{50}$) increase of 5.2% compared to the coastal region with a median temperature ($T_{50}$) increase of 3.6%.

Across the entire study area and throughout almost all seasons, we predominantly observe higher MMP during El Niño phases, as compared to non-ENSO conditions (Figure 4 and Table S 2). The exception is the summer season, in which roughly two thirds (58.7%) of the river catchments experience reduced mean monthly precipitation. The catchments displaying decreased MMP during the summer season are predominantly located in the humid-temperate and semi-arid regions and show MMP decreases of up to -35.0% ($P_5$, Table S 3). On a yearly basis, the highest increase in precipitation is found in the semi-arid

region (median: +32.3%), followed by the mediterranean region (median: +28.4%) and the humid-temperate region (median: +10.8%). The highest precipitation increase in the humid-temperate region occurs during spring.

Overall, it seems that the increase in mean monthly precipitation is higher in the Andes compared to the coastal region. However, this may be because Andean stations are predominantly located in the semi-arid and mediterranean regions, whereas coastal region catchments dominate in the humid-temperate region. The precipitation and temperature anomalies that occur

during El Niño events, based on the full 1961-2009 dataset, are shown in the data supplement (Figure S 8).

#### 4.1.2. Differences in mean monthly specific discharge (MMQsp) and discharge variability

During El Niño events, the MMQsp are higher in the majority of the catchments (96.0%, Table S 2) as compared to non-ENSO periods (Figure 4). The increases in specific discharge are evident across all seasons, but the largest increase is found in summer with a median change of +56.5% (Table S 3). The relative increase in specific discharge is highest in the semi-arid region

(median: +104.4%), followed by the mediterranean region (median: +40.1%) and the humid-temperate region (median: +23.9%). Interestingly, the magnitude of specific discharge increase is higher than the increase in MMP. For all stations and all seasons, the Kolmogorov-Smirnov test revealed significant differences in the daily specific discharge distributions between El Niño and non-ENSO phases.



The observed expansions in the quantile area of the high-flow regime ($\Delta QA_{95}$) reveal significant increases in the frequency
and/or the magnitude of high flows during El Niño phases, compared to non-ENSO conditions, particularly in Andean
catchments located in the semi-arid and mediterranean climate zones (Figure 5; see Figure S 10 for individual station data).
This observation holds across all seasons, but the increases in the high flow quantile area are less extreme in the spring season.
Catchments in the humid-temperate region display only minor differences in the quantile area of high flows on an annual basis,
but $\Delta QA_{95}$ increases more noticeably during the autumn and summer seasons (Figure 5, see Figure S 10 for individual station
data). Even though there is some scatter, the differences in the quantile area of low flows ($\Delta QA_5$) show predominantly negative
values, which reveals that low flows occur less often during El Niño phases, as compared to non-ENSO phases (Figure S 10).
The results from the inverse gamma fitting reveal both increased and decreased discharge variability, as indicated by the
colour-coding in Figure 5 (see Figure S 11Figure S 10 for individual station data). On an annual basis, we find a decrease in
discharge variability during El Niño events (blue markers), despite the above-described increases in the quantile area of the
high-flow regime (Figure 5). This reveals that, on an annual basis, the frequency and/or the magnitude of the high discharges
increase significantly with El Niño, but the frequency of intermediate discharges increases even more, so that overall, the
variability of discharge is reduced (i.e., even though they increase, extreme discharges deviate less from the mean, because the
latter increases further) (Figure S 7c). However, on a seasonal basis we also observe increasing discharge variability
(decreasing k-value, red colours). This occurs in some parts of the semi-arid region and the mediterranean regions during
winter and spring, and in some parts of the humid-temperate region during autumn, spring and summer. This reveals that in
these regions and for these seasons, El Niño has affected the high-flow regime more strongly than the low and intermediate
flow regimes. Because the high-flow regime deviates more strongly from the mean, the discharge variability has increased.

### 4.2. La Niña

#### 4.2.1. Differences in mean monthly temperature (MMT) and mean monthly precipitation (MMP)

During La Niña events, about two third of the catchments (70.5%) experience higher mean monthly temperatures (Figure 6
and Table S 4) compared to non-ENSO phases. There is a latitudinal contrast in the regions that predominantly experience
increasing versus decreasing temperatures. In the semi-arid region in the north, temperatures are lower in the majority of the
catchments (74.1%); half of the catchments show decreasing temperatures in the mediterranean region (50.0%), whereas in
the humid-temperate region in the south the temperatures are higher in almost all (97.1%) of the catchments (Table S 4). The
highest temperature anomalies are found in the semi-arid region, where temperature decreases by up to -382.6% ($T_5$) and
increases by up to +69.8% ($T_{95}$) (Table S 5). The temperature increase in the humid-temperate region is highest during the
autumn and spring seasons, and lowest during the summer season (Figure 6 and Table S 5). Over the entire study area, MMP
rates are lower in most of the catchments (95.5%) during La Niña phases compared to non-ENSO periods (Figure 6 and Table
S 4). As observed with El Niño conditions, one exception in the typical pattern is the summer season, during which 35.3% of
the catchments experience higher MMP (Table S 4). The catchments with increased MMP in summer are mainly located in
the north (semi-arid and mediterranean climate zones) (Figure 6). The decreases in mean monthly precipitation are highest
during the autumn (median: -47.0%) and spring (median: -34.8%) seasons (Figure 6 and Table S 5). The decrease in MMP
shows a latitudinal pattern, being highest in the semi-arid region (median: -33.2%), followed by the mediterranean (median: -
31.0%), and lowest in the humid-temperate (median: -22.2%) regions. The precipitation and temperature anomalies during La
Niña events, based on the full 1961-2009 data, are shown in the data supplement (Figure S 12).

#### 4.2.2. Differences in mean monthly specific discharge (MMQsp) and discharge variability

The MMQsp is lower in most of the catchments (89.5%) during La Niña phases, as compared to non-ENSO conditions, a
signal that is persistent throughout all seasons (Figure 6 and Table S 4). The largest decrease in specific discharge is found in
the autumn (median: -36.4%) and spring (median: -30.5%) seasons (Table S 5). The reduction in specific discharge is largest




in the semi-arid region (median: -55.7%) in the north, whereas the mediterranean and humid-temperate regions show almost similar decreases in specific discharge (median: -19.3% and -20.5%, respectively) (Table S 5).

Despite some scatter, the quantile areas for high flows ($\Delta QA_{95}$) predominantly decrease during La Niña events as compared to non-ENSO phases (Figure S 12). This pattern is persistent over all seasons except for the summer, in which some catchments show increases in $\Delta QA_{95}$. The quantile area for low flows ($\Delta QA_5$) is higher during La Niña events as compared to non-ENSO

phases, especially in the humid-temperate region (Figure 7, see Figure S 12 for individual station data). This reveals that low-flow events either occur more frequently or show a decrease in magnitude, or both. This pattern is persistent over the autumn, spring, and summer seasons. For the winter season a roughly opposite latitudinal pattern is visible. The quantile area of low flow is higher in the semi-arid region and lower in the humid-temperate region, relative to the other seasons. The inverse gamma fitting results reveal overall decreasing discharge variability (cool marker colours) over almost the entire latitudinal

extent in the Andes, except for some regions during the autumn (Figure 7, see Figure S 13 for individual station data). Discharge variability increases (warm marker colours) more often for many catchments in the coastal region, except for the spring season, when it predominantly decreases (Figure 7, see Figure S 13 for individual station data).

## 5. Discussion

### 5.1. Temperature and precipitation anomalies during El Niño and La Niña

Previous studies that investigated ENSO-induced climatic anomalies in this part of the world typically present El Niño in central Chile as the warm and wet phase, while La Niña is described as the cold and dry phase (Diaz and Kiladis, 1995; Garreaud et al., 2009; Hernandez et al., 2022; Jaksic, 1998; Montecinos et al., 2000) – a pattern linked to sea surface temperatures anomalies in the Southern Pacific Ocean (Díaz and Kiladis, 1995; see Section 4.2). Overall, our study confirms the spatial and seasonal patterns in temperature and precipitation anomalies found in previous studies, and provides additional

insight on differences between de coastal region and the Andes (Garreaud et al., 2009; Garreaud and Battisti, 1999; Hernandez et al., 2022; Meza, 2013; Montecinos et al., 2000; Montecinos and Aceituno, 2003; Oertel et al., 2020). Mean monthly temperatures are slightly higher in most of the catchments during El Niño events, except for some catchments in the semi-arid region which experience colder temperatures (Figure 4 and Table S 2). The warmer air temperatures during El Niño phases have been attributed to the heating of air over the warmer ocean (Díaz and Kiladis, 1995). Garreaud et al. (2009) also reported

cooler temperatures in the semi-arid region during El Niño, which they attributed to the higher precipitation and cloud cover, that reduce direct solar insolation and increase surface moisture. Hernandez et al. (2022) found that cold late spring temperatures in the same region and speculated besides the higher frequency of cloudy days whether the above average precipitation could increase the Bowen ratio leading to increasing evaporative cooling and a lengthening of the snow season. Although the nature of our data does not allow us to determine whether these are indeed the driving mechanisms, we find that

the region with the strongest precipitation increases overlaps with the region with colder temperatures. During La Niña phases, temperatures decrease in the semi-arid region and in half of the catchments of the mediterranean region, but increase in the humid-temperate region in the south, as compared to non-ENSO conditions (Figure 6 and Table S 4). This finding of higher temperatures in the south contradicts with typically cooler conditions that are associated with La Niña events in central Chile, driven by cold sea surface temperatures (Diaz and Kiladis, 1995). However, La Niña phases cannot be considered as exactly

the opposite of the warm and wet El Niño conditions, as pointed out by Montecinos et al. (2000). Even though there are spatial and seasonal variations in MMT anomalies during El Niño and La Niña years, a seasonal pattern that can, like MMP, be attributed to the position or intensity of the SPH, is absent. Potentially, the abnormal behaviour of the SPH during the summer season, as compared to the rest of the year, could be the reason why temperature anomalies in summer deviate from temperature anomalies in the other seasons, but this needs to be further investigated.



The largest relative changes in precipitation during both El Niño and La Niña phases are found in the semi-arid region, followed by the mediterranean region, and then the humid-temperate region (Figure 4 and Figure 6). We suggest that this stems from the fact that an absolute increase or decrease of a few millimetres of MMP in a semi-arid region results in a large relative change, as compared to a change of a few millimetres in a more humid region (Garreaud et al., 2009; Hernandez et al., 2022). The wetter-than-normal conditions that we report during El Niño agree with previous studies and can be linked to a weakening

of the intensity of the SPH, whereas the drier-than-normal conditions during La Niña can be linked to a strengthening of the intensity of the SPH (Garreaud and Battisti, 1999; Hernandez et al., 2022; Montecinos and Aceituno, 2003; Rutllant et al., 2003). The seasonal variation in the latitudinal position of the strongest climatic anomalies has been attributed to the seasonal movement in the position of the SPH (Montecinos and Aceituno, 2003). During both El Niño and La Niña events, the precipitation anomalies during the summer season deviate from those in other seasons (Figure 4 and Figure 6). This has been

observed in previous studies and is related to the contrasting summer behaviour of the SPH, as compared to the other seasons (Garreaud et al., 2009; Montecinos et al., 2000; Montecinos and Aceituno, 2003). The anomalously dry conditions in the summer season during El Niño phases have been linked to the intensification of ridges in the southern tip of the SPH, which are also responsible for the arid conditions in northern Chile (Montecinos and Aceituno, 2003).

### 5.2. Specific discharge anomalies during El Niño and La Niña

Mean monthly values of specific discharge (MMQsp) follow the expectations of wet conditions during El Niño and dry conditions during La Niña, but we observe distinct spatial and seasonal patterns (Figure 4 and Figure 6). Following the MMP anomalies pattern, the largest relative differences in MMQsp during El Niño and La Niña phases are found in the semi-arid region, followed by the mediterranean region, and then the humid-temperate region (Table S 3 and Table S 5). As was discussed for the MMP anomalies (Section 5.1), this can be explained by the fact that a change of similar absolute magnitude has a larger

relative effect in a semi-arid region as compared to a humid region.

The ENSO-induced hydrological anomalies during both El Niño and La Niña events are most extreme in the Andean catchments, compared to catchments from the coastal region. The Andes feature higher elevation and steeper topography compared to the coastal region and are therefore characterised by lower temperatures, sparser vegetation cover (Alvarez-Garreton et al., 2018), and likely thinner regolith thicknesses. All these factors impose reduced evapotranspiration and

infiltration rates, which most likely lead to less modulated, flashier river discharge responses to precipitation events in the Andes, when compared to the coastal region. However, it is not possible to assess whether these contrasting catchment attributes affect the observed differences in hydrological anomalies between the coastal region and the Andes, because both regions are subjected to different climatic forcing in the first place.

Even though the annual patterns of MAP and MAQsp anomalies during the El Niño phase look very similar, the seasonal

MMQsp patterns are quite different from the corresponding MMP patterns, suggesting a non-linear relationship between precipitation and river discharge (Figure 4). Direct comparison between ΔMMP and ΔMMQsp during El Niño events reveals that the seasonal deviations between precipitation and river discharge anomalies are largest in the Andes as compared to the coastal region (Figure 8). Furthermore, the MMP and MMQ anomalies deviate most strongly in the summer and autumn seasons, as reflected by the large scatter, but show smaller deviations during the winter season. We explain these observations

by the time lag between precipitation input and river discharge output, introduced by inter-seasonal hydrological storage within each catchment. At such timescale, water can naturally be stored in a catchment mostly in the form of snow accumulation, groundwater storage, or storage in the unsaturated zone of regolith or fractured bedrock. We suggest that the higher precipitation input during El Niño autumn and winter in the northern Andean catchments will partly enhance streamflow (Figure 4) but also increase snow accumulation, causing a delayed hydrological response during the snowmelt season, as

indicated by the strong increases in mean discharge and high flow during summer. This is supported by previous studies that measured enhanced snow accumulation in the Andes during El Niño events (Cordero et al., 2019; Cortés and Margulis, 2017;




Masiokas et al., 2006; Oertel et al., 2020). Other studies reported a similar river discharge response in northern-Andean basins during El Niño spring and summer and also attributed this to an enhanced snowmelt peak (Hernandez et al., 2022; Piechota et al., 1995; Waylen et al., 1993; Waylen and Caviedes, 1990). Finally, Masiokas et al. (2006) directly compared ENSO-induced

snow accumulation to streamflow patterns and found highly significant correlations and similar interannual fluctuations between mean snow water equivalent in the Andes and river discharge of 10 Chilean and Argentinean rivers between 31-37°S. However, as briefly mentioned above, a fraction of the winter precipitation in northern-Andean basins also produces direct streamflow. This can be observed by strong increases in MMQsp (Figure 4), the increase in the high-flow quantile area (Figure 5), and the extreme outliers in the ΔMMP-ΔMMQsp comparison plot (Figure 8). An increase in winter streamflow in high-

elevation Andean basins has also been observed in previous studies and has been attributed to El Niño-induced warm precipitation (i.e., rainfall) and rain-on-snow events, which produce high magnitude river discharge (Waylen et al., 1993; Waylen and Caviedes, 1990).

Catchments that are located in the humid-temperate region receive the highest precipitation anomalies during spring, due to the seasonal southward shift of the SPH (Montecinos and Aceituno, 2003). Where catchments have thick regolith, enhanced

precipitation likely infiltrates the regolith and recharges groundwater storage, which then provides baseflow in the summer and autumn seasons, observed by lower MMP anomalies but higher MMQ values. The observed increase in high flows (increasing $\Delta QA_{95}$) during El Niño may be partially explained by a two times higher likelihood of intense precipitation events (Poveda et al., 2020). However, the enhanced snow accumulation, groundwater storage and wetter antecedent soil moisture due to a longer storm duration (Hernandez et al., 2022) can additionally contribute to an increase in high flows during the

snowmelt and low-flow season, as their contribution to baseflow provides an additional discharge contribution on top of already wet El Niño conditions (Figure 5).

Despite those few cases where the increase in the quantile area of high flow ($\Delta QA_{95}$) overlaps with increasing discharge variability (e.g., winter in the semi-arid region), we observe that for Andean catchments both on an annual basis and for most seasons, an increase in $\Delta QA_{95}$ is combined with decreasing discharge variability. This reveals that during El Niño, the daily

discharge distribution often shifts towards higher discharge magnitudes, but the maximum magnitude of high flows does not significantly increase (Figure S 8c). The decrease in discharge variability likely results from the fact that enhanced snowmelt constitutes a shift of the discharge distributions towards higher discharge magnitudes. Less variable discharge is typical for snow-covered basins (Deal et al., 2018; Rossi et al., 2016; Scherler et al., 2017; Waylen et al., 1993), because snowmelt produces non-flashy river discharge over a longer hydrological response time. The fact that an increase in the quantile area of

high flow ($\Delta QA_{95}$) during El Niño in coastal region catchments often pairs with increasing discharge variability suggests that such high-flow events are due to intense precipitation storms (Poveda et al., 2020) or large precipitation storms with a long storm duration (Hernandez et al., 2022). To conclude, during El Niño events, both the mean river discharge and high-flow regime predominantly increase in the semi-arid and the mediterranean regions. Andean snow cover is found to be an important hydrological process that introduces seasonal modulation of the ENSO-induced climatic anomalies.

During La Niña events, both the annual and seasonal MMQsp anomalies show a similar spatial pattern when compared to the MMP anomalies (Figure 6). The decrease in both MMP and MMQsp is strongest in the semi-arid region, followed by the mediterranean and humid-temperate regions. Strikingly, at the annual scale, the shift of the discharge distribution towards low-flow magnitudes during La Niña phases predominantly occurs in the humid-temperate region (Figure 7), which indicates that the large reductions in mean discharge in the semi-arid and mediterranean regions are predominantly driven by decreases in

the intermediate to high discharge magnitudes. This is also explained by decreasing discharge variability ($\Delta k$, blue markers) in Andean catchments (Figure 7). We interpret this to be caused by the presence of snow cover and glaciers in the semi-arid and mediterranean regions, which maintain the low-flow regime, even during periods of low precipitation (Masiokas et al., 2006; Milana, 1998). This may partly result in runoff directly generated from snow or glacier melt, but also by groundwater discharge, which are all mechanisms generating streamflow at a later stage (Alvarez-Garreton et al., 2021; Ayala et al., 2020).



When La Niña events follow upon El Niño events, part of the El Niño-enhanced snow accumulation may still persist, especially in the case of high-elevation catchments (Cordero et al., 2019; Cortés and Margulis, 2017; Masiokas et al., 2006). Furthermore, as snowmelt is an important source for groundwater recharge and snow-dominated catchments have a longer hydrological memory, higher groundwater levels resulting from enhanced snow accumulation during El Niño can provide higher baseflow during subsequent La Niña events (Alvarez-Garreton et al., 2021). We also suggest that this snowmelt contribution to river

discharge in Andean catchments is the reason why discharge variability generally decreases during La Niña (Deal et al., 2018; Rossi et al., 2016; Waylen et al., 1993). Because snow cover and glaciers are less abundant in the humid-temperate region, the hydrological regime is of pluvial type (Alvarez-Garreton et al., 2021; Oertel et al., 2020). Snowmelt cannot maintain a minimum baseflow in this region, which makes it more sensitive to precipitation deficits during La Niña events.

### 5.3. Implications for water resources management

This study reveals that ENSO-induced hydrological anomalies vary strongly in magnitude and seasonality across the various climatic zones and in between the Andes and the coastal region. This underlines the importance of high-resolution observational studies that assess regional variabilities of catchments hydrological responses to climatic anomalies, as well as the need for region-specific water resources and risk management strategies (Ayala et al., 2020; Blöschl et al., 2019; Kemter et al., 2020).

As observed for other regions in the world (Lee et al., 2018; Mosley, 2000), snow cover plays a crucial role in modulating the ENSO-induced climatic anomalies in Andean catchments in central Chile. Not only do the high precipitation anomalies during El Niño winters drive a direct, rainfall-induced increase in discharges in the same winter season, but furthermore, the enhanced snow accumulation causes a second, snowmelt-generated increase in river discharge in the subsequent summer (Piechota et al., 1995; Waylen et al., 1993; Waylen and Caviedes, 1990). This specifically affects the semi-arid and mediterranean regions,

which experience MMQsp increases of up to +397% and +105% ($Q_{95}$, Table S 3), respectively, as well as strong increases in the high-flow regime (Figure 4 and Figure 5). In fact, the semi-arid and mediterranean regions are known as flood-risk prone areas, with reports of large floods coinciding with El Niño events in the past (Aceituno et al., 2009; Jenny et al., 2002; Waylen and Poveda, 2002).

Despite its influence on flood risk, the enhanced snow cover typically resulting from El Niño events is also crucial for water

availability in the semi-arid and mediterranean regions (Cordero et al., 2019; Masiokas et al., 2006). Snowmelt-generated runoff has the potential to reduce the impact of droughts (during, e.g., subsequent La Niña events) in these regions, because water from snow and glacier melt contribute to a minimum discharge level, as discussed in Section 5.2 (Masiokas et al., 2006; Milana, 1998).

The effects of ENSO on snow accumulation and river discharge, therefore, present major socioeconomic challenges for the

semi-arid and mediterranean regions in central Chile, where ~55% of the population of the country resides. On the one hand, enhanced flooding risks during El Niño events constitute a significant threat to the population and infrastructure (Ward et al., 2014, 2016). On the other, the predominant source for crop irrigation, domestic water use, and hydropower generation in these regions is river discharge from snow and glacier melt (Alvarez-Garreton et al., 2018; Cordero et al., 2019; Masiokas et al., 2006). Even though the precipitation deficits that incur during La Niña periods affect the hydrology of the humid-temperate

region most strongly, as shown by a higher frequency of low-flow events, these presumably have a smaller socioeconomic impact, as this region generally receives higher rainfall, has larger groundwater storages, and the current water demands for crop irrigation and hydropower are much smaller, in relative terms, as compared to the mediterranean and semi-arid part of the country (Alvarez-Garreton et al., 2018; Masiokas et al., 2006).

Incorporating probabilistic ENSO flood and drought risk forecasting on top of regular meteorological-forcing forecast, could

improve flood and drought predictions. However, the strong modulation of ENSO-induced climatic anomalies shows that flood and drought risk assessments cannot be derived directly from ENSO-induced precipitation anomalies, but should also consider



other hydrologically-relevant processes (Emerton et al., 2017, 2019; De Perez et al., 2017), such as snow cover, groundwater fluctuations, and antecedent hydrological conditions (e.g., Alvarez-Garreton et al., 2021; Hernandez et al., 2022; Masiokas et al., 2006). Considering the lead time of ENSO forecasts, and the subsequent time lag between snow accumulation and melt, there is potential for early flood and drought warning and mitigation in snow-covered basins.

Furthermore, water resources management strategies are required to mitigate increasing water-related challenges for central Chile under future climate change, as modelling studies predict that the climate will shift towards more arid conditions (Boisier et al., 2018a; Cai et al., 2020). Moreover, as the zero-degree isotherm is expected to rise in Andean basins (Mardones and Garreaud, 2020), already expressed by rapidly shrinking Andean glaciers (Barcaza et al., 2017; Braun et al., 2019; Dussaillant et al., 2019), it is possible that currently snowmelt-dominated basins will shift towards a rainfall-dominated discharge regime in the future, and thus shrinking the forecast time. How the frequency and amplitude of ENSO events will respond to future climate change is debated. Stevenson (2012) compared results from different climate models and observed an increase in ENSO amplitude in only 4 out of 11 models, whereas other studies suggest a future increase in the frequency of El Niño events (Cai et al., 2014). Under the latter scenario, the semi-arid and mediterranean regions may experience wetter-than-normal conditions more frequently, which is advantageous under the predicted expanding arid conditions. However, due to the above-described expected transition of snowfall-dominated towards rainfall-controlled hydrology, the strong increases in winter precipitation would increase the risk of winter floods. Therefore, on top of drought-mitigation strategies for an overall more-arid future climate, flood risk management may become increasingly important in the semi-arid and mediterranean regions, to mitigate El Niño-induced winter floods.

No matter whether or not the amplitude and frequency of ENSO events Increase under climate change, it is clear that ENSO-induced climatic anomalies and the resulting inter-seasonal hydrological anomalies will undoubtedly add extra complexity to climate adaptation strategies. Modelling studies on the changes in climate and ENSO amplitude and frequency, and the specific effects they would have on central Chile are needed to better constrain future water-related challenges in central Chile.

### 6. Conclusion

In this study, we investigated the effects of the El Niño Southern Oscillation (ENSO) on climatic and hydrological anomalies in 178 river catchments located in central Chile. Generally, we observed increasing precipitation and specific discharge rates during El Niño and decreasing precipitation and discharge rates during La Niña, but we detected large spatial and seasonal variations. The semi-arid region experiences the strongest climatic and hydrological anomalies during both El Niño and La Niña phases, followed by the mediterranean and the humid-temperate regions. Furthermore, we conclude that rain- and snow-dominated catchments display a contrasting hydrological response to ENSO, as snow cover in the Andes strongly modulate ENSO-induced climatic anomalies, whereas rainfall-dominated basins show a discharge response that is somewhat modulated by groundwater, but overall, show larger similarities with ENSO-induced climatic anomalies.

During El Niño, river catchments located in the semi-arid and mediterranean regions are found to experience high-flow conditions during winter, when the climatic anomalies are highest, but also during summer as a result of a delayed discharge peak induced by enhanced snowmelt. River discharge from rainfall-dominated basins in the humid-temperate region is most strongly affected by precipitation deficits during La Niña, whereas snowmelt-generated runoff provides a minimum low flow in snowmelt-dominated basins.

These ENSO-induced climatic and hydrological anomalies add extra complexity on top of an already challenging climate change scenario for central Chile. This poses major socio-economic challenges, in particular for the semi-arid and mediterranean regions of Chile, where the majority of the population resides, that are strongly dependent on river water for hydropower generation, crop irrigation, and domestic use. We conclude that improved probabilistic ENSO flood and drought risk forecasts and water resources management strategies are required to mitigate floods and droughts under future climate



change: These strategies should be performed differently for snow- and rainfall-dominated catchments, due to their different responses to ENSO.

### 7. Dataset availability

All supplementary tables (S1-S5) and figures (S1-S14) are available in the data supplement.

The CAMELS-CL dataset was downloaded from the World Data Center PANGAEA (https://doi.pangaea.de/10.1594/PANGAEA.894885). The CR2MET precipitation and temperature products were downloaded from the CR2 website (http://www.cr2.cl/datos-productos-grillados). The WorldClim V2.1 historical monthly precipitation data was downloaded from the WorldClim website (https://www.worldclim.org/data/monthlywth.html). The Global Precipittaion Climatology Centre (GPCC) mean monthly precipitation dataset was downloaded from the data portal of the Deutschee Wetterdienst (https://opendata.dwd.de/climate_environment/GPCC/html/fulldata-monthly_v2022_doi_download.html). The mean monthly precipitation dataset from the Climate Hazards Group InfraRed Precipitation with Station data (CHIRPS) was downloaded from the Columbia Climate School Internatione Research institute for Climate and Society (https://iridl.ldeo.columbia.edu/SOURCES/.UCSB/.CHIRPS/.v2p0/.monthly/.global/.precipitation/). The Climate Prediction Center mean monthly temperature dataset was downloaded from the Columbia Climate School Internatione Research institute for Climate and Society (https://iridl.ldeo.columbia.edu/SOURCES/.NOAA/.NCEP/.CPC/.GHCN_CAMS/.gridded/.deg0p5/.temp/index.html). The Shuttle Radar Topography Mission Global 90m digital elevation data (SRTM GL3) was downloaded from OpenTopography (https://portal.opentopography.org/raster?opentopoID=OTSRTM.042013.4326.1).

### 8. Author contributions

R. van Dongen carried out the data analysis with help from all co-authors. D. Scherler acquired the project funding and was the main supervisor throughout the study. D. Wendi conducted the interpolation of the data gaps in the daily discharge time series. C. Meier supported the data interpretation in the context of Chilean climate and hydrology. R. van Dongen prepared the manuscript with contributions from all co-authors.

### 9. Competing interests

The authors declare that they have no conflict of interest.

### 10. Acknowledgements

We acknowledge support from the German Science Foundation (DFG) priority research programme SPP-1803 "EarthShape: Earth Surface Shaping by Biota" (grant SCHE 1676/4-1 to D.S.). We thank K. Übernickel, F. von Blanckenburg, and T. Ehlers for support and coordinating the EarthShape SPP.



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



## 12. Figures

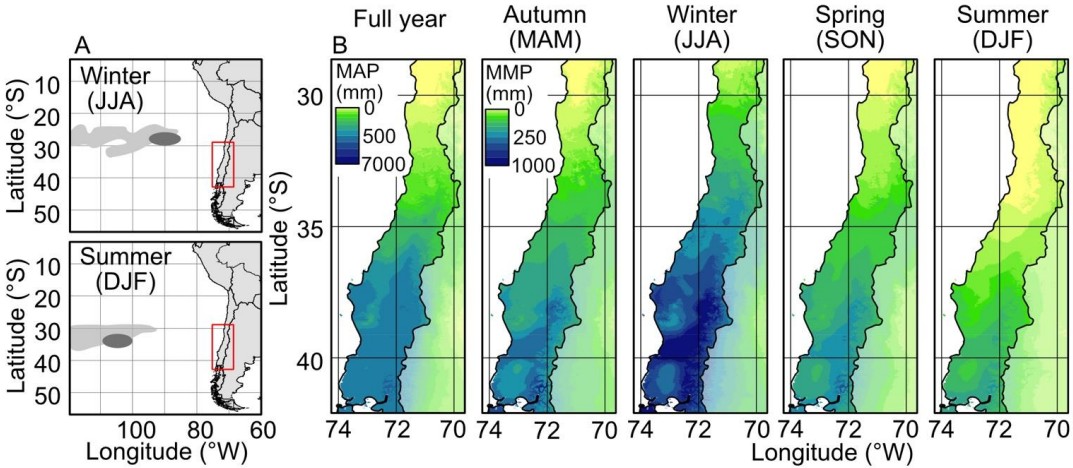


**Figure 1: Seasonal variations of the Southern Pacific High (SPH) location and precipitation rates. A) Location of the Southern Pacific High anticyclone in the winter and summer seasons according to the studies of Barrett and Hameed (2017) (light grey area, SPH locations between 1980-2013) and Schneider et al. (2017) (dark grey spot). The red boxes indicate the extent of the research area. B) Mean annual precipitation (MAP) and seasonal variations in mean monthly precipitation (MMP) in central Chile. The**
**colourmaps have a logarithmic scale.**



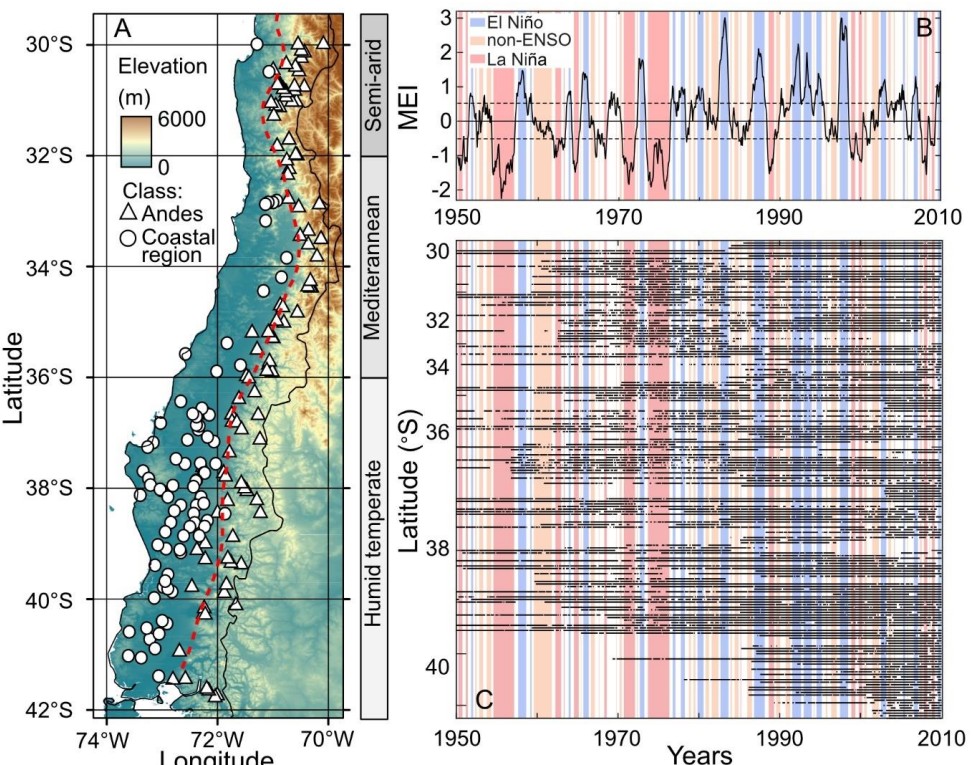

**Figure 2: Locations of river discharge station and data coverage in comparison to Multivariate ENSO Index (MEI) values. A) 90m resolution Digital Elevation Model of central Chile (Jarvis et al., 2008), with the locations of the river discharge stations in the Andes (triangles) and coastal region (circles). On the right side, the extent of the climatic zones: semi-arid (29-32°S), mediterranean (32-36°S), and humid-temperate (36-42°S). B) Multivariate ENSO index (MEI; Wolter and Timlin, 1993, 1998, 2011) with the classified El Niño (blue), La Niña (red) and non-ENSO (yellow) events. C) Data coverage of each river discharge station over the time period 1950-2009.**






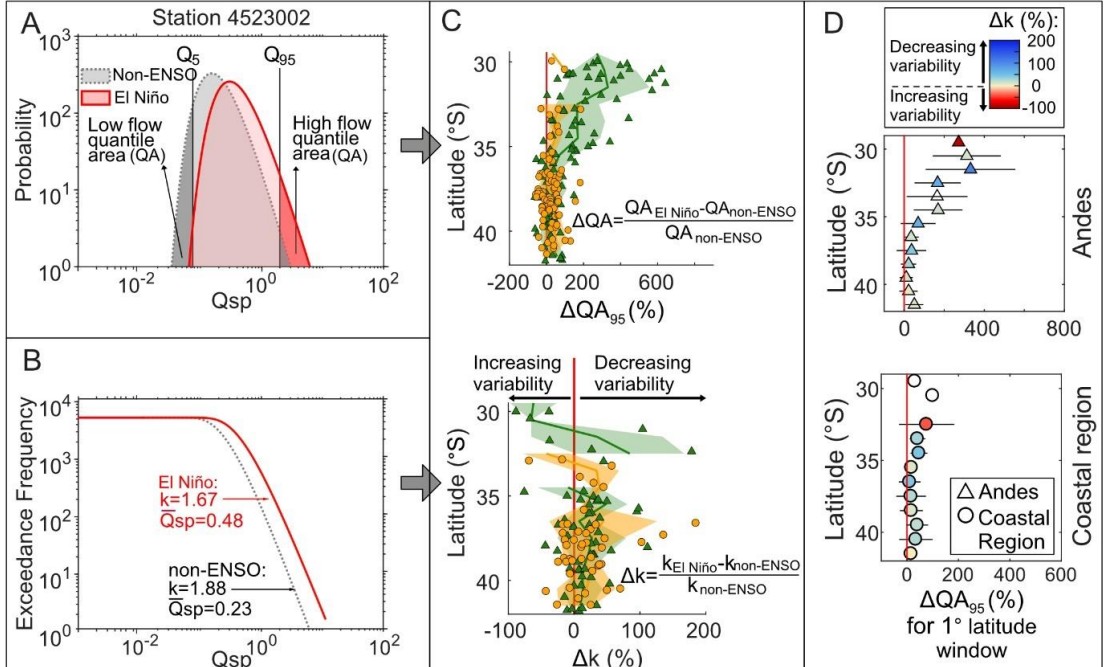

**Figure 3: Description of the ΔQA and inverse gamma fitting methods, to investigate the differences in the high-flow and low-flow regimes and discharge variability. This example is based on a comparison between El Niño and non-ENSO and focusses on the high-flow regime in panels C and D (QA95). A) Example of an empirical probability distribution for station 4523002_Río Grande en Puntilla San Juan (Table S 1). The figure shows the empirical daily specific discharge distributions (Qsp) for El Niño (red) and non-ENSO (grey). The areas above and below the 5th- and 95th-percentiles of non-ENSO events (Q5 and Q95) represent the low flow quantile area (QA5) and high flow quantile area (QA95), respectively. B) The exceedance frequency distribution of daily specific discharge data (Qsp) for the El Niño (red) and non-ENSO (grey) phases is parameterized using inverse gamma fitting (Lague et al., 2005). High discharge variability is characterized by a heavy tailed distribution and a low k-parameter, whereas low discharge variability is characterized by a high k-parameter. Only inverse gamma fits with sufficient data (>3 months of data) and a good fit (r2>0.97) are used for further analysis. C) For all 178 stations, the differences in low flow and high flow quantile areas (ΔQA5 and ΔQA95, upper panel) and discharge variability (Δk, lower panel) are calculated as percent difference of El Niño relative to the non-ENSO phase. The resulting ΔQA and Δk values plotted separately for Andean stations (green triangles) and coastal region stations (orange circles). Furthermore, a latitudinal average (solid line) and 1σ standard deviation (shaded background) is calculated for all stations within a 1°-latitudinal window. D) Finally, the results of both methods are combined in one figure. The markers indicate the mean ΔQA and the error bars the 1σ-standard deviation for all stations in each 1° latitude window. The symbols are colour-coded by the value of Δk. Red values indicate increasing discharge variability and blue values decreasing discharge variability, based on the inverse gamma fit.**





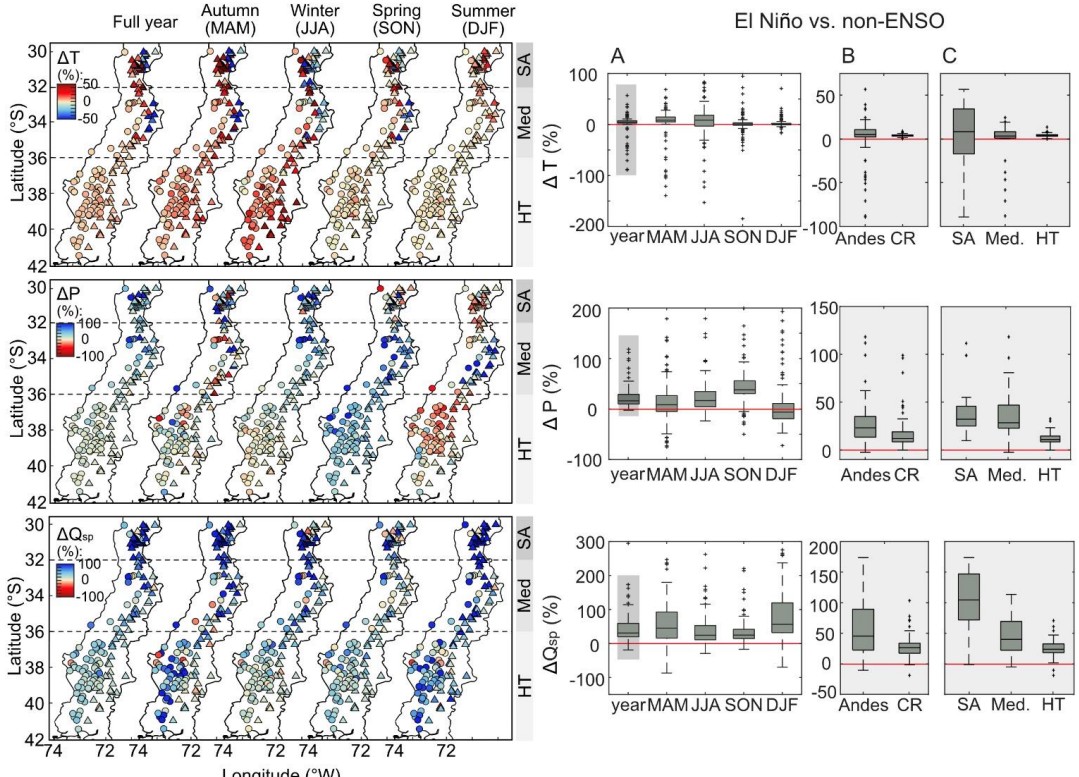

**Figure 4: Differences (in %) in mean monthly temperature (ΔT), precipitation (ΔP), and specific discharge (ΔQsp) during El Niño events relative to non-ENSO conditions. The maps on the left show the spatial and seasonal differences for the full year and the different seasons. The dotted lines indicate the boundaries between the different climate zones, which are indicated in the subpanels on the right: semi-arid (SA), mediterranean (Med), and humid-temperate (HT). The boxplots in column A) show the variation of the data for the full year and the individual seasons. The grey shaded area represents the y-axis extent of the boxplot columns B) and C) that are based on the annual data. Boxplot column B) shows the differences between the Andes and the coastal region (CR), while boxplot column C) shows the differences between the three climate zones. The precipitation and temperature anomalies for the full 1961-2009 period are shown in the data supplement (Figure S 9).**



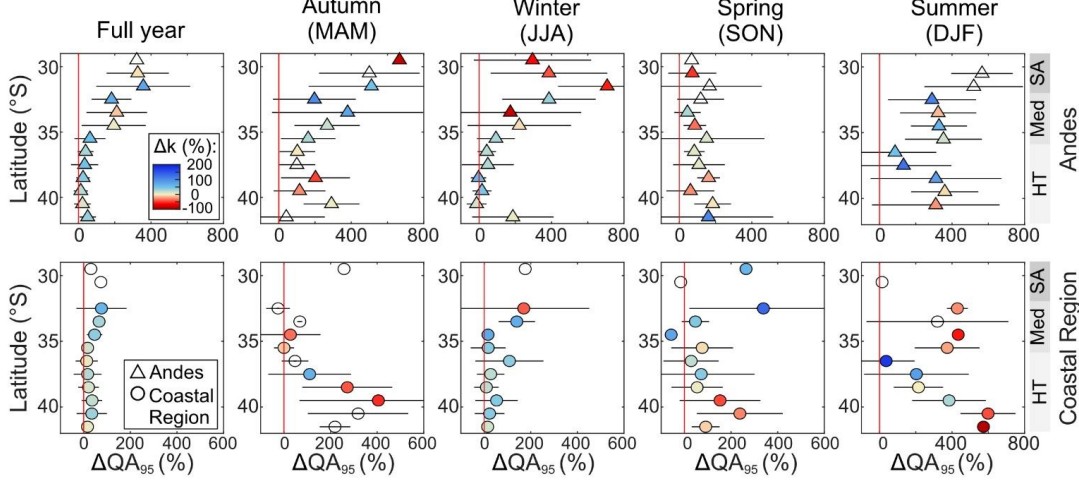

**Figure 5: Latitudinal (1°) averages of the percent differences in the quantile area (ΔQA) and k-parameter (Δk) when comparing the high-flow regime during El Niño events to non-ENSO conditions. See Figure S 10 and Figure S 11 for individual station results for ΔQA and Δk. The method to create this figure is described in Figure 3. The markers present the mean value, and the error bars the 1σ-standard deviation, for all stations located in each 1°-latitudinal window. The upper row presents the changes in high-flow regime for the Andean region, while the lower row shows the changes for the coastal region. The markers are colour-coded by the difference in k-parameter between El Niño and non-ENSO periods. Redder markers reveal increasing discharge variability, while bluer markers reflect decreasing discharge variability, according to the inverse gamma fitting. When the marker colour is missing, no high-quality (R2>0.97) fit exists for the latitudinal window.**





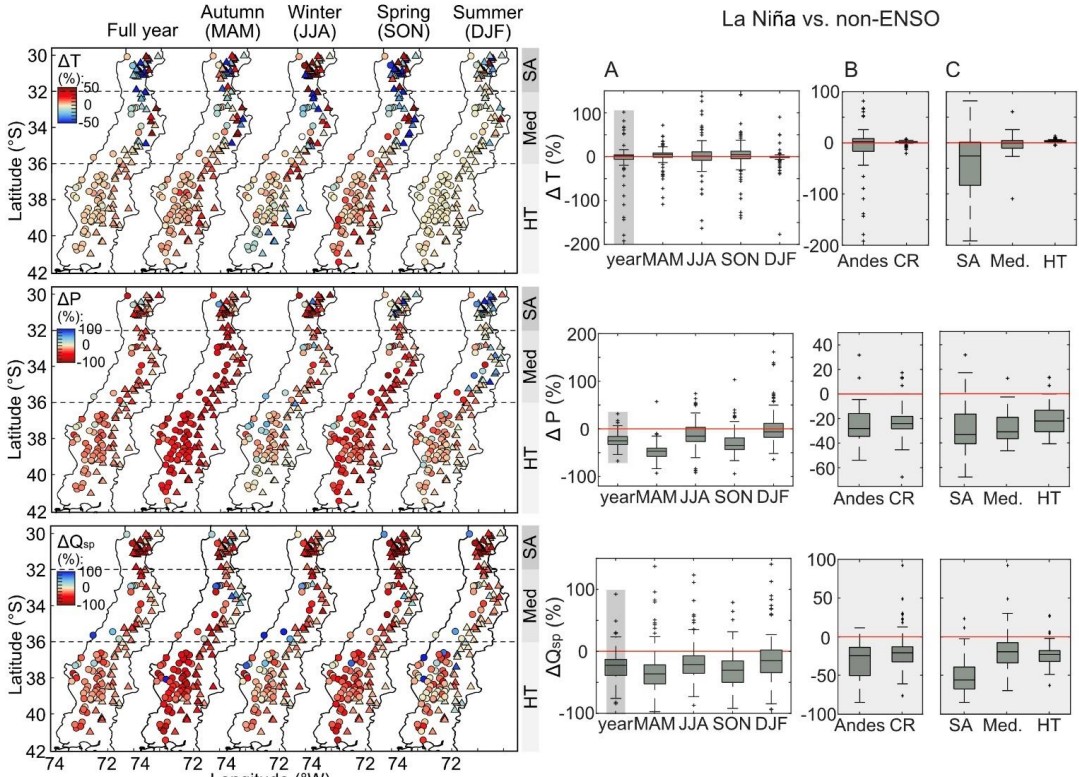

**Figure 6: Differences (in %) in mean monthly temperature (ΔT), precipitation (ΔP), and specific discharge (ΔQsp) during El Niño events relative to non-ENSO conditions. The maps on the left show the spatial and seasonal differences for the full year and the different seasons. The dotted lines indicate the boundaries between the different climate zones, which are indicated in the subpanels**
970 **on the right: semi-arid (SA), mediterranean (Med), and humid-temperate (HT). The boxplots in column A) show the variation of the data for the full year and the individual seasons. The grey shaded area represents the y-axis extent of the boxplot columns B) and C) that are based on the annual data. Boxplot column B) shows the differences between the Andes and the coastal region (CR), while boxplot column C) shows the differences between the three climate zones. The precipitation and temperature anomalies for the full 1961-2009 period are shown in the data supplement (Figure S 12).**

975





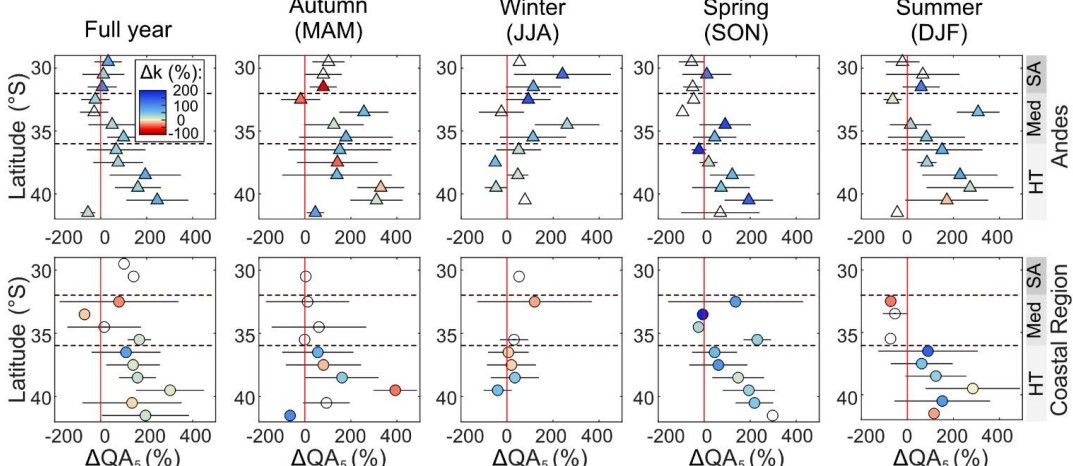

**Figure 7: Latitudinal (1°) averages of the differences in the quantile area of the low-flow regime during La Niña events compared to non-ENSO conditions. See Figure S 13 and Figure S 14 for individual station results of ΔQA and Δk. The method to create this figure has been described in Figure 3. ΔQA and Δk present the percent differences of La Niña events relative to non-ENSO conditions. The markers present the mean value and the error bars the 1σ-standard deviation of all stations in each 1°-latitudinal window. The upper row presents the changes in high-flow regime for the Andean region, the lower row the changes for the coastal region. The markers are colour-coded by the difference in k-parameter between La Niña and non-ENSO periods. Red markers reveal increasing discharge variability and blue markers reveal decreasing discharge variability according to the inverse gamma fitting. When the marker colour is missing, no high-quality (R2>0.97) fit exists for the latitudinal window.**



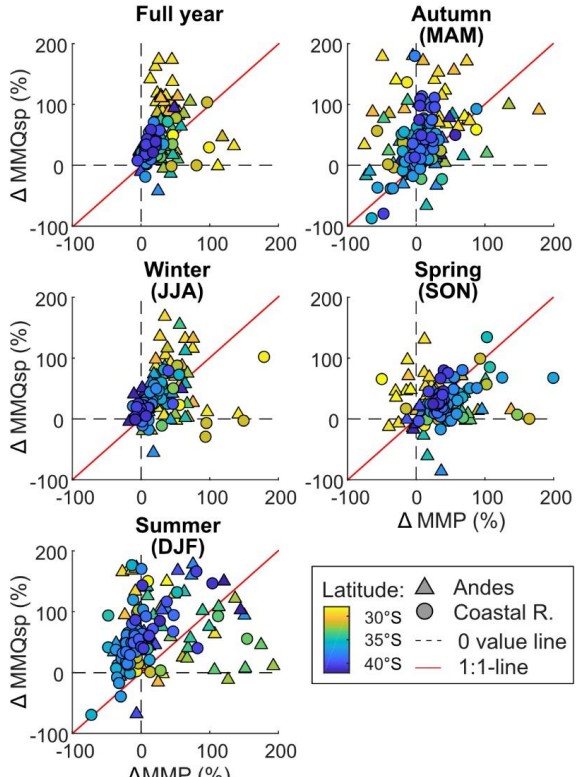

990

**Figure 8: Scatterplot comparing differences in mean monthly precipitation (MMP) and mean monthly specific discharge (MMQsp) between El Niño and non-ENSO conditions. The scatter includes basins located in the Andes (triangles) and the coastal region (dots), the colour-coding represents the latitude. The dotted lines indicate the boundaries between the positive and negative domain. Red line is the 1:1-line.**