# Peer review of "El Niño Southern Oscillation (ENSO)-induced hydrological anomalies in central Chile"

_EGUsphere, 2022_

## Referee Comment (RC1)

El Niño Southern Oscillation (ENSO) – induced hydrological anomalies in central Chile

Authors: R. van Dongen, D. Scherler, D. Wendi, E. Deal, L. Mao, N. Marwan, C. Meir

**1.- General comment**

In my opinion this manuscript does not add significant new information to the well documented knowledge of ENSO impacts on rainfall, surface temperature and river discharge regimes in central Chile. In addition to this, many errors and deficiencies in the presentation, some of them quite basic, led me to conclude that in its present version this paper does not reach the minimum standard of quality needed for publication.

**2.- Specific comments**

**2.1.- Lines 185 – 186: Data selection**: "…*we selected stations based on data availability during the chosen time period (1961 – 2009), requiring a record length of at least 10 years"*. 178 catchments listed in Table S1 were selected based on this and other requirements.

Observation:

Considering the main purpose of the study, which is to characterize hydrological anomalies during El Niño and La Niña events, a minimum record length of 10 years is too small to achieve this goal with results that are statistically significant. Furthermore, the list of 178 stations in Table S1 includes 20 stations with record lengths below 10 years during the time period selected for the analysis (1961 – 2009), with some of them as short as 3 years (stations 6000003, 7317003, 7317005, 8117006, 8312001, 8313000, 8372002, 10100006, 10122003, 10327001, 10351001, 10401001, 10405002, 10431000, 10432003, 10503001, 10514001, 10520001, 10523002).

**2.2.- Lines 201 – 202: Classification of river catchments**: "… *we classified the river catchments into Andean (high elevation) or coastal region (low elevation) river catchments*"

Observation:

This classification is quite arbitrary and ignores the main characteristic of topography of central Chile from around 33°S to 40°S, dominated by the Andes cordillera on the east where, on the average, precipitation occurs as snowfall during winter in areas above 3.000 m above sea level and by the central valley and a coastal range where precipitation occurs mostly as rainfall. Thus, just a few of the so-called coastal stations corresponds to low-level coastal basins where precipitation always occurs as rainfall. In most of the stations identified as coastal, located in the middle or lower part of Andean basins, the ENSO impact on river discharge is a mixture of a simultaneous impact on rainfall during the rainy season and a delayed impact on snow melting in the high Andes during summer.

Furthermore, there are a large degree of redundancy among some of the selected hydrological time series, corresponding to measurements of the same river at different elevation within its basin (for example stations 8123001, 8124001, 8124002, 8133001, 8135002 of the Itata river; stations 8307002, 8312000, 8312001, 8317001, 8319001, 8334001 of the river Bio Bio). In these cases, differences between records at different elevation in the basin could prove more useful to separate the ENSO impact on river discharge in the Andes and in the central valley.

Station Rio Hurtado en Entrada Embalse Recoleta (N° 4506002), with mean elevation of 2264.4 m, is classified as "Coastal Region", which is clearly a mistake.

**2.3.- Line 208 – 214: Filling missing data in daily hydrological records**

Observations:

A more clear explanation is needed of the method that was used, particularly regarding the determination of the "tolerable gap lengths" for each station. Presumably gap lengths were shorter than 10 days, as all months containing data gaps longer that 10 days were removed, as indicated in line 210.

Regarding In the example presented in Fig. S2 it is unclear which stations are considered in this figure. There is a reference to 516 river discharge stations across Chile (line 176). From these 516 stations 178 of them were selected for further analysis (line 184), based on several requirements, but it is not mentioned how many stations were initially considered for the region of the study (29° - 42°S). I suppose these are the nearly 320 stations included in Fig. S2

**2.4.- References to previous studies**

Observations:

The manuscript includes a large number of references to previous investigations about ENSO-related climate (rainfall and temperature) and hydrological anomalies in central Chile and also about the physical mechanisms involved. But some statements regarding those references are wrong or incomplete:

Lines 143 – 145: *"… El Niño conditions develops from pressure differences above the Pacific ocean that weaken or reverse the equatorial trade wind, pushing warm sea surface waters from the Western Pacific ocean toward the coast of South America"*. This statement ignores the weakening of the equatorial upwelling and the reduced difference in the sea level along the equatorial Pacific as major factors for the positive sea surface temperature anomalies during El Niño episodes.

Lines 146 – 147: "*Due to the anomalously warm sea surface temperature near the coast of South America, this state is also termed as the warm phase of an ENSO event*". This is incorrect. El Niño is identified as the warm phase of ENSO in association with the positive sea surface temperature anomalies that occur during El Niño episodes along the central and eastern equatorial Pacific.

Lines 151 – 152: "*During El Niño phases the SPH intensity weaken, which results in the blocking of storm tracks across the Admunsen-Bellinghausen Sea and the intensification of the westerlies at mid-latitudes…*". The weakening of the SPH is not the cause of the establishment of a blocking high pressure system over the Admunsen-Bellinghausen Sea during El Niño episode. This is part of a teleconnection pattern triggered by the enhanced atmospheric convection in the central Pacific. Furthermore, the weakening of the SPH in subtropical latitudes combined with the blocking high pressure system in the south explain a weakening of the westerlies at mid-latitudes

Lines 159 – 160: "*When ENSO condition prevail, either El Niño or La Niña, the SPH changes its intensity in the summer season*". This statement is meaningless. In fact throughout the entire year

El Niño episodes are associated with a weaker than normal SPH while the opposite occurs during La Niña events.

**2.5.- **Comparison between CR2MET and WorldClim V2.1 rainfall data sets** (lines 233 – 235)**

Observation:

It is mentioned that compared to the CR2MET data set, the WorldClim v2.1 data set was found to overestimate precipitation during the rainy season from April to September (lines 234 – 235).  What it is shown in Fig. S4 is just the opposite, with MMP CR2MET values exceeding by a factor larger than 2.0 the MMP WorldClim values. This is recognized in line 241. Furthermore no information is given regarding the regional characteristics of the bias of these rainfall estimations when compared with rainfall measured at meteorological stations. Apparently it is considered that the accuracy of the CR2MET data set is acceptable everywhere.

**2.6.- **Comparison between CPC and CR2MET mean monthly surface temperature data** sets (lines 236 – 240)**

Observation:

The comparison is made in Fig. S5 at a monthly scale considering all stations in the semi-arid, mediterranean and humid-temperate regions. This figure shows the existence of large differences between the two data sets, with the CPC estimations underestimating the CR2Met ones by values as large as -10.0 °C at individual stations. Fig. S5 also shows that the difference is larger at stations in the Andes, so the large difference of -5.95°C  that is documented year-round in line 240 for the semi-arid region is explained by the fact that most of the stations considered for this regions are in the Andes (Fig. S1H)

**2.7.- Adoption of the WorldClim v2.1 rainfall and the CPC surface temperature data sets**

Observation:

In spite of the fact that the WorldClim v2.1 rainfall and the CPC surface temperature data underestimate the supposedly closer to reality CR2MET data, in some stations by a large amount, the World Clim and CPC data sets were chosen in the study. This decision is not questionable if the interannual variability in the two data sets of temperature and rainfall is similar, but this is not verified in the article.

**2.8.- **Period used for the analysis**

Observation:

The chosen time period for the analysis is 1961-2009. According to this, it is strange that some results are presented for the period 1950 – 2010 (Fig. 2b,c and Figs. S6 and S7). This discrepancy led to the error in lines 268 - 269 where in reference to Fig. S6 it is mentioned that over the 1961-2009 time period, non-ENSO periods are relatively evenly distributed.

**2.9.- Differences in temperature, rainfall and river discharge during El Niño and La Niña episodes with respect to neutral ENSO conditions** (Figs. 4 and 6).

Observations:

Figures 4 and 6 summarize the differences in rainfall, temperature and river discharge when El Niño and La Niña conditions prevail in the central Pacific, with respect to values observed during neutral ENSO conditions (wrongly named in the article as non-ENSO conditions), at the annual and seasonal time scales for each station (left panels); considering all stations all together (panels in column A); for stations in the Andes and those labeled as "coastal region" (panels in column B), and at a regional scale considering all the stations within the semi-arid, mediterranean and humid-temperate regions (panels in column C). I have several observations regarding the way the results are presented:

a) In my judgement, the most serious deficiency in this article is the lack of a rigorous assessment of the statistical significance of the differences that are presented. So, it is impossible to discriminate which of the differences presented in Figs. 4 and 6 as well in Tables S3 and S5 may have occurred by chance or were determined by the occurrence of El Niño or La Niña episodes.

b) Mean river discharge differences with respect to neutral ENSO conditions at individual stations during El Niño and La Niña episodes are not comparable between them, even at nearby stations, due to the different record length of the time series (see Fig. 2c).

c) The methodology used to calculate rainfall differences during the summer season is useless for the semi-arid and most of the mediterranean region, where it does not rain during this season.

d) Differences in temperature expressed as percentage during El Niño and La Niña episodes with respect to neutral ENSO conditions is not standard and hard to interpret in physical terms (¿how many °C correspond to the maximum value of +253.24% indicated in Table S3 for the $T_{95}$ in the semi-arid region ?)

e) Differences at the seasonal scale of river discharge do not consider its seasonal delay in the response associated to snow melting during the spring and summer. In fact, regarding ENSO impacts on river discharges, particularly in the semi-arid and mediterranean regions, the maximum values in the annual cycle occurring during summer (DJF) are mostly conditioned by the ENSO state during the previous rainy season in winter. So, for these two regions at least the impacts of El Niño on river discharge in summer, when it is reached the maximum in the annual cycle, should be calculated with a delay of 6 months considering the occurrence of El Niño conditions during the previous winter.

f) Usefulness of results presented in column A of Figs. 4 and 5 are doubtful as they ignore the latitudinal and altitudinal differences in the ENSO impacts on temperature, rainfall and river discharge.

g) Results presented in panels of column B ignore the uneven relative distribution of "Andes" and "CR" stations in the three latitudinal regions. In particular, in the semi-arid zone most of the station are Andean, while in the humid-temperate zone most of the stations are classified as "coastal", as shown in Fig. 2.

---

## Author Comment (AC1)

**Authors response**

**Title:** El Niño Southern Oscillation (ENSO)-induced hydrological anomalies in central Chile

**Authors:** Renee van Dongen, Dirk Scherler, Dadiyorto Wendi, Eric Deal, Luca Mao, Norbert Marwan, Claudio I. Meier

**Manuscripts code:** egusphere-2022-1234

We thank Cristian Chadwick and anonymous reviewers 1 and 2 for their time and effort in reviewing this manuscript. We appreciate the constructive feedback of all reviewers, which will significantly improve the manuscript and, therefore, we agree with improving the paper based on a number of suggestions raised by all reviewers.

Reviewer 1 raised concerns that our study does not add new knowledge and reviewer 2 found the novel aspects of our new results not well represented in the discussion and conclusion. Although it's true that a number of studies have investigated the effect of ENSO on temperature and precipitation anomalies in central Chile, our main purpose of including temperature and precipitation data is to present the forcing factors of anomalies in mean discharge and the frequency and magnitude of low and high flow events. We believe that the novelty of this study lies in the detailed attention at the effect of ENSO on the hydrological response (in particular the magnitude and frequency of low and high flow events, using the quantile area and inverse gamma fitting methods) and the implications for water resources management. It appears that the reviews focused most of their attention to the analysis and results at the beginning of the paper (i.e. the mean differences in temperature, precipitation and river discharge), and we thus decided to restructure the paper in order to highlight our novel contribution in the field of effects on magnitude and frequency of flow events.

To date, a couple of studies have also investigated the hydrological response to ENSO in central Chile (e.g. Hernandez et al., 2022; Oertel et al., 2020, Piechota et al., 1995; Rubio-Álvarez and McPhee, 2010; Waylen et al., 1993; Waylen and Caviedes, 1990, Yan Yan et al., 2020). However, these studies were either global studies based on model results (Yan Yan et al., 2020), or included only a few (Andean) gauging stations over a long ~1200 km stretch (Oertel et al., 2020, Piechota et al., 1995; Rubio-Álvarez and McPhee, 2010; Waylen et al., 1993; Waylen and Caviedes, 1990). The most recent study by Hernandez et al. (2020) included 59 gauging stations, but these were also predominantly located in the Andes, and this study was conducted with an annual rather than a seasonal focus and did not investigate hydrological extremes. Therefore, we believe that our manuscript represents a significant contribution as it is based on a large dataset (178 stations) of monitored time series of river discharge stations located both in the Andes and in the coastal range. The expansion of these areas is important to capture and understand the regional spatial variation of such a distinctive, yet connected, topography gradient. Furthermore, we believe that due to the complex hydrological processes taking place in central Chile, it is highly relevant to focus on the seasonal patterns and shifts, the occurrence of hydrological extremes, and to discuss the implications for water resources management. Having said that, we thank the reviewers for making us realize that the manuscript needs to be restructured in order to communicate clearer our original contribution.

The lack of statistical analysis was identified by reviewer 1 as the most serious shortcoming of this study. Although we did perform a statistical analysis (as described in lines 376-378), which resulted in significant differences in all cases, we did not include the analyses in a figure. However, we agree that the results of the tests should have been communicated much more clearly and we will re-evaluate the statistical analysis and report this clearly in the revised manuscript.

Below we reply in greater detail on the specific comments in the reviews of:

    A)  Community comment Cristian Chadwick
    B)  Anonymous reviewer 1
    C)  Anonymous reviewer 2

After considering these explanations and commitments to revise the manuscript (also in the specific responses to each reviewer below), we hope that the Editor will consider this study for a potential publication.

**A) Community comment Cristian Chadwick**

**General Comments:**

1) In the results section, the authors estimate the changes in temperature, as relative changes. To me, this makes no sense, because it is highly influenced by the baseline temperature. For example, if the baseline temperature is 10°C and one has a temperature increase of 1°C (between two different phases of ENSO), the percentage of change is a 10%, but if the baseline temperature was 0.5°C a 1°C increase would lead to a 200% change. In an even more extreme case if the baseline temperature of was -0.5°C a 1°C increase, would lead to a -200% change. To avoid, these types of problems I would recommend one of two options: Option 1) to use absolute temperature changes, or Option 2) to use relative changes, but with Kelvin degrees, which avoid all the issues beforementioned. Some of the results presented, I think might be influenced by using relative changes in temperature, and might mislead the reader of your article.

*Thank you for this suggestion. In the revised manuscript version, we will use absolute Kelvin degrees for temperature to avoid these problems.*

2) I would add an analysis of a trends, especially for temperature. If there is a temperature trend in the study zone and period, and given that most of the La Niña years have happened before 1980, that might bias your temperature analysis for "La Niña", in case you detect temperatures increasing in time.

*Good point, we will add this analysis to the revised manuscript. We carried out a preliminary trend analysis for the temperature, precipitation and river discharge data, using the Mann-Kenndall for the entire time period (1961-2009). This showed no significant trends for the precipitation and river discharge data and a significant positive trend for the temperature data for 32 stations. We will extend this analysis for different periods where certain ENSO-phases dominate and discuss the implications in the revised manuscript.*

**Minor Comments 1-5:**

*We will improve the manuscript based on all minor suggestions kindly provided by the reviewer.*

**B) Reviewer 1**

**2.1.- Lines 185 – 186: Data selection**: "…*we selected stations based on data availability during the chosen time period (1961 – 2009), requiring a record length of at least 10 years".* 178 catchments listed in Table S1 were selected based on this and other requirements.

Observation:
Considering the main purpose of the study, which is to characterize hydrological anomalies during El Niño and La Niña events, a minimum record length of 10 years is too small to achieve this goal with results that are statistically significant. Furthermore, the list of 178 stations in Table S1 includes 20 stations with record lengths below 10 years during the time period selected for the analysis (1961 – 2009), with some of them as short as 3 years (stations 6000003, 7317003, 7317005, 8117006, 8312001, 8313000, 8372002, 10100006, 10122003, 10327001, 10351001, 10401001, 10405002, 10431000, 10432003, 10503001, 10514001, 10520001, 10523002).

*Authors' response:*
*We agree that a longer record length would be of course desirable for this study, but this would result in the loss of a large number of stations. The choice of a minimum record length of 10 years was a reasonable compromise between the loss of stations by increasing the minimum record length and what we considered to be a minimum requirement for our analysis. Based on this comment we now realized that a couple of stations which were included in the dataset had a data duration of less than 10 years. This occurred when we shortened the time period after the decision to exclude the mega drought (all years from 2010 onwards). We will amend this and improve the description of the rationale behind our choice of stations and length of the analysis period in the revised manuscript.*

**2.2.- Lines 201 – 202: Classification of river catchments**: "*… we classified the river catchments into Andean (high elevation) or coastal region (low elevation) river catchments*"

Observation:
This classification is quite arbitrary and ignores the main characteristic of topography of central Chile from around 33°S to 40°S, dominated by the Andes cordillera on the east where, on the average, precipitation occurs as snowfall during winter in areas above 3.000 m above sea level and by the central valley and a coastal range where precipitation occurs mostly as rainfall. Thus, just a few of the so-called coastal stations corresponds to low-level coastal basins where precipitation always occurs as rainfall. In most of the stations identified as coastal, located in the middle or lower part of Andean basins, the ENSO impact on river discharge is a mixture of a simultaneous impact on rainfall during the rainy season and a delayed impact on snow melting in the high Andes during summer.

Furthermore, there are a large degree of redundancy among some of the selected hydrological time series, corresponding to measurements of the same river at different elevation within its basin (for example stations 8123001, 8124001, 8124002, 8133001, 8135002 of the Itata river; stations 8307002, 8312000, 8312001, 8317001, 8319001, 8334001 of the river Bio Bio). In these cases, differences between records at different elevation in the basin could prove more useful to separate the ENSO impact on river discharge in the Andes and in the central valley. Station Rio Hurtado en Entrada Embalse Recoleta (N° 4506002), with mean elevation of 2264.4 m, is classified as "Coastal Region", which is clearly a mistake.

*Authors' response:*
*Thanks for this suggestion, we will change the catchment classification to two or three classes (rainfall dominated, snowmelt dominated and potentially a mixed class) based on monthly Pardé coefficients for normalized streamflow as input for the K-means clustering method (a similar method as described in Hernandez et al., 2022). Furthermore, we will include a discussion of the implications of nested catchments in the discussion. We agree that the classification of station 4506002 went wrong with our original classification procedure, but it will be reclassified in the new system in the revised manuscript.*

**2.3.- Line 208 – 214: Filling missing data in daily hydrological records**

Observations:
A more clear explanation is needed of the method that was used, particularly regarding the determination of the "tolerable gap lengths" for each station. Presumably gap lengths were shorter than 10 days, as all months containing data gaps longer that 10 days were removed, as indicated in line 210.

Regarding In the example presented in Fig. S2 it is unclear which stations are considered in this figure. There is a reference to 516 river discharge stations across Chile (line 176). From these 516 stations 178 of them were selected for further analysis (line 184), based on several requirements, but it is not mentioned how many stations were initially considered for the region of the study (29° - 42°S). I suppose these are the nearly 320 stations included in Fig. S2.

*Authors' response:*
*As we described in the manuscript (lines 210-212) each station had its specific tolerable gap length, based on the statistical characteristics, i.e., the acceptable maximum days of lag-autocorrelation to determine this tolerable gap days. This tolerable gap length differs from station to station, but was in all cases below 10 days, the maximum threshold. We will better explain this statistical method in the revised manuscript. Reviewer 1 is right that the CAMELS-CL dataset includes 516 stations over entire Chile (17.8°S-55.0°S), the ~320 stations were indeed all the stations within our study range (29°S-42°S) before removing stations based on data availability and other catchment criteria. We will modify Fig. S2 to only show the stations used in this figure.*

**2.4.- References to previous studies**
Observations:
The manuscript includes a large number of references to previous investigations about ENSO-related climate (rainfall and temperature) and hydrological anomalies in central Chile and also about the physical mechanisms involved. But some statements regarding those references are wrong or incomplete:

Lines 143 – 145: *"… El Niño conditions develops from pressure differences above the Pacific ocean that weaken or reverse the equatorial trade wind, pushing warm sea surface waters from the Western Pacific ocean toward the coast of South America"*. This statement ignores the weakening of the equatorial upwelling and the reduced difference in the sea level along the equatorial Pacific as major factors for the positive sea surface temperature anomalies during El Niño episodes.

Lines 146 – 147: "*Due to the anomalously warm sea surface temperature near the coast of South America, this state is also termed as the warm phase of an ENSO event*". This is incorrect. El Niño is identified as the warm phase of ENSO in association with the positive sea surface

temperature anomalies that occur during El Niño episodes along the central and eastern equatorial Pacific.

Lines 151 – 152: "*During El Niño phases the SPH intensity weaken, which results in the blocking of storm tracks across the Admunsen-Bellinghausen Sea and the intensification of the westerlies at mid-latitudes…*". The weakening of the SPH is not the cause of the establishment of a blocking high pressure system over the Admunsen-Bellinghausen Sea during El Niño episode. This is part of a teleconnection pattern triggered by the enhanced atmospheric convection in the central Pacific. Furthermore, the weakening of the SPH in subtropical latitudes combined with the blocking high pressure system in the south explain a weakening of the westerlies at mid-latitudes.

Lines 159 – 160: "*When ENSO condition prevail, either El Niño or La Niña, the SPH changes its intensity in the summer season*". This statement is meaningless. In fact throughout the entire year El Niño episodes are associated with a weaker than normal SPH while the opposite occurs during La Niña events.

*Authors' response:*
*Thanks for making us aware of the shortcomings in §2. We regret that some details got lost or were wrongly summarized in the descriptions of the complex ENSO mechanisms as a background chapter. We will improve these aspects in the revised manuscript version.*

**2.5.- Comparison between CR2MET and WorldClim V2.1 rainfall data sets (lines 233 – 235)**

Observation:
It is mentioned that compared to the CR2MET data set, the WorldClim v2.1 data set was found to overestimate precipitation during the rainy season from April to September (lines 234 – 235). What it is shown in Fig. S4 is just the opposite, with MMP CR2MET values exceeding by a factor larger than 2.0 the MMP WorldClim values. This is recognized in line 241. Furthermore no information is given regarding the regional characteristics of the bias of these rainfall estimations when compared with rainfall measured at meteorological stations. Apparently it is considered that the accuracy of the CR2MET data set is acceptable everywhere.

*Authors' response:*
*Apologies for the mistake in lines 234-235 and many thanks for spotting it! We will correct this in the revised manuscript.*
*We considered the CR2MET to be the best performing dataset because in a quick comparison between basin average precipitation and basin average discharge (calculated from the CAMELS-CL dataset), the CR2MET dataset showed an expected behavior and the least scatter (see Fig. 1). Furthermore, the good performance of the CR2MET dataset was also communicated by Chilean hydrologists (personal communication: René Garreaud, Center for Climate and Resilience Research). This dataset performs especially well, because, different from globally gridded data products, it is specifically developed for Chile, it has a smaller resolution (0.05°), and is quality-checked with a large dataset with about 7 million daily precipitation observations from 866 precipitation stations. A report by the Dirección General de Aguas (DGA) about the development of the CR2MET dataset, revealed a weaker performance in the far north (<20°S) and in Patagonia (>45°S), but reported high $R^2$ values (0.7-1) between the gridded product and ground observations for our study area (29°S-42°S) (e.g., Figure 5.10, DGA, 2017). We will include this information and cite this report in the revised manuscript.*

[Figure]

*Figure 1: Comparison between basin average Mean Monthly Precipitation calculated from the CR2MET gridded dataset (MMP$_{CR2MET}$) and the Mean Monthly specific discharge (MMQ$_{sp}$) calculated from the CAMELS-CL data.*

**2.6.- **Comparison between CPC and CR2MET mean monthly surface temperature data sets**
(lines 236 – 240)

Observation:
The comparison is made in Fig. S5 at a monthly scale considering all stations in the semi-arid, mediterranean and humid-temperate regions. This figure shows the existence of large differences between the two data sets, with the CPC estimations underestimating the CR2Met ones by values as large as -10.0 °C at individual stations. Fig. S5 also shows that the difference is larger at stations in the Andes, so the large difference of -5.95°C that is documented year-round in line 240 for the semi-arid region is explained by the fact that most of the stations considered for this regions are in the Andes (Fig. S1H)

*Authors' response:*
*By re-evaluating available monthly gridded temperature products, we have found additional reanalysis datasets that provide temperature data covering our study period:*

- *NOAA-CIRES 20th Century Reanalysis (V2)*
- *NOAA-CIRES 20th Century Reanalysis (V2c)*
- *NOAA/CIRES/DOE 20th Century Reanalysis (V3)*
- *U. of Delaware Precipitation and Air Temperature*
- *NCEP-NCAR Reanalysis 1*

*As with the precipitation datasets, we will compare their quality with the CR2MET temperature data and select the best dataset for our revised manuscript. In addition, we would like to emphasize that we have presented the temperature and precipitation data mainly to show the forcing factors affecting river discharge, but that the main focus of this study is the river discharge data.*

**2.7.- **Adoption of the WorldClim v2.1 rainfall and the CPC surface temperature data sets**

Observation:
In spite of the fact that the WorldClim v2.1 rainfall and the CPC surface temperature data underestimate the supposedly closer to reality CR2MET data, in some stations by a large amount, the World Clim and CPC data sets were chosen in the study. This decision is not questionable if

the interannual variability in the two data sets of temperature and rainfall is similar, but this is not verified in the article.

*Authors' response:*
*That's a good point, we will perform this analysis and discuss it in the revised manuscript.*

2.8.- **Period used for the analysis**

Observation:
The chosen time period for the analysis is 1961-2009. According to this, it is strange that some results are presented for the period 1950 – 2010 (Fig. 2b,c and Figs. S6 and S7). This discrepancy led to the error in lines 268 - 269 where in reference to Fig. S6 it is mentioned that over the 1961-2009 time period, non-ENSO periods are relatively evenly distributed.

*Authors' response:*
*Originally, we performed the study for the time period 1950-2009 using another precipitation dataset, but due to the poor correlation of monthly P the CR2MET dataset we decided to use the WorldClim v2.1 dataset which does not provide data before 1961. We agree that we should have updated the time series of some of the figures and will do this in the revised manuscript.*

2.9.- **Differences in temperature, rainfall and river discharge during El Niño and La Niña episodes with respect to neutral ENSO conditions** (Figs. 4 and 6).

Observations:
Figures 4 and 6 summarize the differences in rainfall, temperature and river discharge when El Niño and La Niña conditions prevail in the central Pacific, with respect to values observed during neutral ENSO conditions (wrongly named in the article as non-ENSO conditions), at the annual and seasonal time scales for each station (left panels); considering all stations all together (panels in column A); for stations in the Andes and those labeled as "coastal region" (panels in column B), and at a regional scale considering all the stations within the semi-arid, mediterranean and humid-temperate regions (panels in column C).

I have several observations regarding the way the results are presented:
a) In my judgement, the most serious deficiency in this article is the lack of a rigorous assessment of the statistical significance of the differences that are presented. So, it is impossible to discriminate which of the differences presented in Figs. 4 and 6 as well in Tables S3 and S5 may have occurred by chance or were determined by the occurrence of El Niño or La Niña episodes.

   *Authors' response:*
   *We ran the Kolmogorov-Smirnov test on the data, as reported in lines 376-378 of the manuscript. This test showed significant differences in all cases and we have therefore not highlighted this in our figures. We agree that this should have been communicated more clearly in the manuscript. We will re-evaluate our statistical analysis and report this much more clearly in our revised manuscript.*

b) Mean river discharge differences with respect to neutral ENSO conditions at individual stations during El Niño and La Niña episodes are not comparable between them, even at nearby stations, due to the different record length of the time series (see Fig. 2c).

*Authors' response:*
*It is true that the length of river discharge time series varies between stations, but that is all we can work with in terms of ground observations. We consider this to be valid because we are looking at broad patterns, where we lump signals from stations in large regions, rather than comparing individual stations amongst each other. Furthermore, the spatial patterns look overall smooth and the fact that the temperature and precipitation patterns look comparable when using the full time period (Figure S9 and S10) or for matching the available river discharge time series only (Figure 4 and 6) further supports that the record lengths seem representative for the full time period. An alternative would be to model river discharge data, but meteorological and hydrological models also have large uncertainties, especially with the complex topography of Chile. We think there is a huge advantage to using ground observations rather than modelled data when studying the complex controls of ENSO on hydrology in Chile. We will discuss this issue and the related uncertainties in the new version of the manuscript.*

c) The methodology used to calculate rainfall differences during the summer season is useless for the semi-arid and most of the mediterranean region, where it does not rain during this season.

*Authors' response:*
*We do not see major issues if the precipitation data for certain stations are (close to) zero in certain seasons. Also, we prefer not to exclude the possibility of sporadic precipitation in these regions during the summer season, which could occur, for example, during the wet phase of El Niño. Finally, not calculating the precipitation difference for certain stations in certain seasons would require a detailed evaluation and discussion of which stations to exclude and which not to exclude, and scientific views could differ widely on this. Therefore, we prefer to leave this as it is and rather offer a deeper discussion as to why there are large magnitude changes (in %) in this season due to the low background value. In addition, as suggested for temperature, we will consider including an extra figure in the data supplement showing the absolute changes in precipitation.*

d) Differences in temperature expressed as percentage during El Niño and La Niña episodes with respect to neutral ENSO conditions is not standard and hard to interpret in physical terms (¿how many °C correspond to the maximum value of +253.24% indicated in Table S3 for the $T_{95}$ in the semi-arid region ?)

*Authors' response:*
*This point was also raised by Cristian Chadwick in community comment 1. We will present the temperature data as absolute differences in Kelvin degrees in the revised manuscript.*

e) Differences at the seasonal scale of river discharge do not consider its seasonal delay in the response associated to snow melting during the spring and summer. In fact, regarding ENSO impacts on river discharges, particularly in the semi-arid and mediterranean regions, the maximum values in the annual cycle occurring during summer (DJF) are mostly conditioned by the ENSO state during the previous rainy season in winter. So, for these two regions at least the impacts of El Niño on river discharge in summer, when it is reached the maximum in the annual cycle, should be calculated with a delay of 6 months considering the occurrence of El Niño conditions during the previous winter.

*Authors' response:*
*We agree with this comment, and indeed we have discussed this in the manuscript. For example, in lines 541-551 we describe that snowmelt dominated basins are not that sensitive to low precipitation input during La Niña, because snowmelt- and groundwater-generated runoff maintains a baseflow. In the manuscript we specifically described them as hydrological mechanisms that produce streamflow at a later stage and can even stem from El Niño-enhanced snow accumulation during the previous year. Another example where we discussed this in the manuscript: lines 489-491 and lines 494-497. Seemingly, this discussion was not highlighted well enough, and we will thus improve this part of the discussion in order to make it clearer.*

f) Usefulness of results presented in column A of Figs. 4 and 5 are doubtful as they ignore the latitudinal and altitudinal differences in the ENSO impacts on temperature, rainfall and river discharge.

*Authors' response:*
*Panels A, B and C have been created to summarize the visual patterns shown in the maps on the left of Figures 4 and 5. We have specifically produced multiple panels to cover the differences between seasons (A), altitudes (B) and climatic regions (C) to recognize the differences between these groups. The maps on the left can be used by readers who wish to observe the complex seasonal, latitudinal and altitudinal variations all at once in one figure.*

g) Results presented in panels of column B ignore the uneven relative distribution of "Andes" and "CR" stations in the three latitudinal regions. In particular, in the semi-arid zone most of the station are Andean, while in the humid-temperate zone most of the stations are classified as "coastal", as shown in Fig. 2.

*Authors' response:*
*We agree, but there is no workaround for the locations of the good quality discharge stations. We have discussed this issue at least once in the manuscript (e.g., lines 368-369). We will improve the discussion on this issue in the revised manuscript.*

**C) Reviewer 2**

- Although the authors provided a lot of detailed results and discussions in different parts of the study domain under different ENSO phases, most texts are only presenting increasing/decreasing consequence of the precipitation/discharge responding to ENSO while lacking sufficient physical explanations for these informative analysis. This requires a major revision of the whole result/discussion sections to provide more in-depth analysis and discussion.

*We thank reviewer 2 for this suggestion, because it motivates us to better support our physical explanations, which we will improve in the revised manuscript.*

- Highly related to the first comment, the conclusion does not add new information about the impacts of ENSO in this region compared to existing studies including those already mentioned in the manuscript. Although I endorse the reconfirmation of the knowledges

using the ground observations, I would still like to see new interesting information which is not shown in the conclusion.

*Thanks for pointing this out. We agree with the reviewer that we need to better communicate the novel aspects of this study and will improve this in the revised manuscript. We have described in more detail what we consider to be the novelty of this study in our general response at the beginning of this authors' response.*

- The definition of the three hydroclimate regimes needs to be better provided. The 178 discharge stations selected for this study needs to be carefully categorized into the three groups with the consideration of their upstream-downstream relations. For example, the downstream rain-dominant catchment could have upstream river flow inputs from the upstream high elevated snow-dominant catchments, making the analysis a bit complicated.

*We will revise our station classification and classify the stations in two or three classes (rainfall-dominated, snowmelt-dominated and mixed) based on monthly Pardé coefficients for normalized streamflow as input for the K-means clustering method. That is a similar approach as described in Hernandez et al. (2022). Furthermore, we will discuss the implications of nested catchments in the discussion.*

- Skipping the period since 2010 (Lines 165-167) for the analysis might not be a good decision which surprises me actually. At least I would encourage the authors to report the analysis including this part. For example, the following reference shows the significant ENSO-flooding relations during a period of 1998-2013 over the study area. In addition, the reference presented a study using hydrological model simulated discharge without considering irrigation and reservoir operations, i.e., only natural hydrological process is considered, which could provide an useful reference for this study on less human-impacted catchments.

  Yan Yan  Huan Wu,  Guojun Gu,  Philip J. Ward  Lifeng Luo  Xiaomeng Li  Zhijun Huang  Jing Tao, 2020, Exploring the ENSO Impact on Basin-Scale Floods Using Hydrological Simulations and TRMM Precipitation, Geophysical research Letters, https://doi.org/10.1029/2020GL089476

  *We thank the reviewer for pointing out the above reference, which we will include as a comparison for semi-natural catchments. We have discussed the question of including the mega-drought amongst all authors and concluded that excluding the mega-drought (post-2010 period) from our analysis is meaningful, because the origin of the drought is not clear yet. For example, Garreaud et al. (2019) explicitly described the mega-drought to cause abnormally dry conditions during the neutral ENSO phase. Including the period after 2010 is thus likely to bias the non-ENSO phase towards drier conditions, which will affect our results. However, we agree that it makes sense to refer to the drought in the discussion of our results again and will do so in the new version of the manuscript.*

- Figure 2. What is the range of the deltaQA and deltaK to be expected? The impact of the non-ENSO phase on the high- and low flow can be impacted by El Niño and La Niña phases, indicating a non-static non-ENSO phase is used as the reference. Would it be problematic, e.g., in understanding the range of deltaQA and deltaK?

*Good point, we will address the expected range of deltaQA and deltaK in the revised manuscript.*

- Figure 2B. It is a bit hard for me to tell which are the non-ENSO events from those of La Nina events because of the colors used.

*Thanks for letting us know. We will change the color of the non-ENSO events in the revised manuscript.*

***References in this authors' response:***

*DGA: Actualización del Balance Hídrico Nacional, SIT No. 417, Ministerio de Obras Públicas, Dirección General de Aguas, División de Estudios y Planificación, Santiago, Chile, Realizado por: Universidad de Chile & Pontificia Universidad Católica de Chile, 2017.*

*Garreaud, R., Boisier, JP., Rondanelli, R., Montecinos, A., Sepúlveda, H. & Veloso-Águila, D. 2019. The Central Chile Mega Drought (2010-2018): A Climate dynamics perspective. International Journal of Climatology. 1-19. https://doi.org/10.1002/joc.6219*

*Hernandez, D., Mendoza, P. A., Boisier, J. P., & Ricchetti, F. (2022). Hydrologic sensitivities and ENSO variability across hydrological regimes in central Chile (28°–41°S). Water Resources Research, 58, e2021WR031860. https://doi.org/10.1029/2021WR031860*

*Oertel, M., Meza, F. J. and Gironás, J.: Observed trends and relationships between ENSO and standardized hydrometeorological drought indices in central Chile, Hydrol. Process., 34(2), 159–174, doi:10.1002/hyp.13596, 2020.*

*Piechota, T. C., Dracup, J. A., Brown, E. F., McMahon, T. and Chiew, F.: South American Streamflow and the Extreme Phases of the Southern Oscillation, in Proceedings of the Eleventh Annual Pacific Climate (PACLIM) Worskhop, edited by C. M. Isaacs and V. L. Tharp, pp. 85–92, California Department of Water Resources., 1995.*

*Rubio-Álvarez, E. and McPhee, J.: Patterns of spatial and temporal variability in streamflow records in south central Chile in the period 1952-2003, Water Resour. Res., 46(5), 1–16, doi:10.1029/2009WR007982, 2010.*

*Waylen, P. R. and Caviedes, C. N.: Annual and seasonal fluctuations of precipitation and streamflow in the Aconcagua river basin, Chile, J. Hydrol., 120, 79–102, 1990.*

*Waylen, P. R., Caviedes, C. N. and Juricic, C.: El Niño-Southern Oscillation and the surface hydrology of chile: A window on the future?, Can. Water Resour. J., 18(4), 425–441, doi:10.4296/cwrj1804425, 1993.*

*Yan Yan, Huan Wu, Guojun Gu, Philip J. Ward Lifeng Luo Xiaomeng Li Zhijun Huang Jing Tao, 2020, Exploring the ENSO Impact on Basin-Scale Floods Using Hydrological Simulations and TRMM Precipitation, Geophysical research Letters, https://doi.org/10.1029/2020GL089476*